# Toward Fine-Grained Domain Knowledge: Curriculum Pseudo-Labeling for Online Test-Time Adaptation

## Abstract

Online Test-Time Adaptation (OTTA) aims to adapt a pre-trained model to unlabeled test instances under domain shift in an online manner, where domain knowledge that the model accumulates from previously observed mini-batches directly affects its predictions on subsequent instances. Most previous OTTA methods exploit domain knowledge at a coarse-grained batch level, which prevents the model from fully absorbing the domain knowledge. To deal with this problem, we propose a novel framework CUrriculum Pseudo-Labeling for Online Test-time adaptation (CUPLOT), which further mines orderly domain knowledge at a fine-grained instance level. Specifically, CUPLOT prepares the arriving batch as a series of curricula based on the modeled relevance of domain knowledge between the model and instances. Then, the model orderly learns the instances with pseudo-labels generated by class prototypes in each curriculum. In this way, the domain knowledge is accumulated in a fine-grained manner through instances of curricula rather than mini-batches, improving the absorption of domain knowledge and the performance of the model. Theoretically, we prove that the curriculum pseudo-labels could enable the model to have a stronger adaptation ability, resulting in a tighter bound of approaching the Bayes optimal classifier on the target domain.

## 1 Introduction

Online Test-Time Adaptation (OTTA), an emerging paradigm, aims to continue to train a pre-trained model with unlabeled instances from a different target domain in an online manner during test time. Due to the difficulty in collecting training samples from the source domain exactly identical to the target domain encountered during testing, the need to adapt the model in the test phase leads to various applications for OTTA techniques, such as medical image analysis (He et al., 2021; Ma et al., 2022), autonomous driving (Volpi et al., 2022; Bahmani et al., 2023), and speech processing (Lin et al., 2022; Kim et al., 2022).

In OTTA, the model can't access previously observed mini-batches, yet it can accumulate domain knowledge, which directly impacts its predictions on subsequent instances. To accomplish the OTTA task, many approaches have been proposed to exploit domain knowledge in unlabeled test instances. Wang et al. (2020); Gong et al. (2022); Mirza et al. (2022); Zhao et al. (2023) modulate the statistics of the batch normalization layer to update domain knowledge of the model when a test mini-batch arrives. Zhang et al. (2022); Jing et al. (2022); Niu et al. (2023); Lee et al. (2024) perform entropy minimization to satisfy the necessary condition to have learned domain knowledge, i.e., more confident predictions on test instances. Iwasawa & Matsuo (2021); Goyal et al. (2022); Shin et al. (2022); Yang et al. (2022); Döbler et al. (2023); Jang et al. (2023); Wang et al. (2023); Sun et al. (2024) focus on generating pseudo-labels for unlabeled test instances to build an empirical risk estimator, enabling the model to absorb domain knowledge in a supervised learning manner.

Intuitively, the more domain knowledge accumulated from each batch, the more beneficial it is for subsequent predictions. However, most previous OTTA methods only exploit domain knowledge at a coarse-grained batch level, limiting the absorption of the domain knowledge from some representative instances. For instance, if the gradient on an instance is more inconsistent with the overall gradient on the batch, its domain knowledge will be diluted or even harm the absorption of domain knowledge

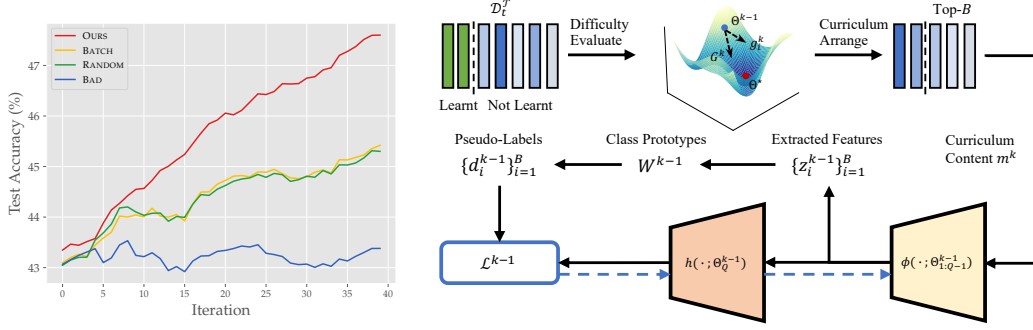

(a) Adaptation accuracy on CIFAR-100-C.   (b) Overview of our approach CUPLOT.

Figure 1: (a) Experimental evidence showing that our gradient-consistency curriculum achieves consistently stronger online adaptation than random, bad, or batch orders on CIFAR-100-C, demonstrating that learning order directly influences the absorption of domain knowledge. (b) Detailed pipeline of the proposed curriculum-driven online adaptation framework at one curriculum.

from other instances, leading to less knowledge being absorbed at the batch level. Empirically, we conduct an additional experiment comparing four learning orders on CIFAR-100-C: (1) our gradient-consistency curriculum order, (2) a random order, (3) an intentionally poor order obtained by reversing the ranked scores, and (4) ordinary batch. As illustrated in Figure 1(a), the adaptation accuracy curves across the test stream show that reasonable ordering consistently achieves higher accuracy than other cases under identical settings, demonstrating that the order in which samples are learned affects how effectively fine-grained domain knowledge is absorbed during online adaptation. Note that without deliberately organizing the learning order of samples, there may even exist learning sequences under which the model almost fails to adapt.

Hence, in this paper, we propose to further mine domain knowledge at a more fine-grained instance level by considering the learning sequence of the instances within each batch from two aspects showed in Figure 1(b). First, the arrived batch is organized as a series of curricula based on the modeled relevance of domain knowledge between what has been learned by the model and what is about to be learned by the model. Second, the model orderly learns the instances with pseudo-labels generated by class prototypes in each curriculum. The proposed framework is named CUPLOT, i.e., *CUrriculum Pseudo-Labels for Online Test-time adaptation*, which accumulates the domain knowledge in a more fine-grained manner through instances of curricula rather than mini-batches, improving the absorption of domain knowledge and the performance of the model. CUPLOT introduces a curriculum mechanism specifically tailored for OTTA by defining easy and hard instances through gradient consistency, a label-free measure reflecting whether a sample's gradient aligns with the dominant update direction. Our contributions are summarized as follows:

- Practically, we propose a curriculum learning framework for OTTA, which prepares the arriving batch as organized curricula and generates pseudo-labels deeply dependent on the curricula, improving the absorption of domain knowledge at a fine-grained level.
- Theoretically, we demonstrate that the curriculum pseudo-labels could enable the model to have a stronger adaptation ability, resulting in a tighter bound of approaching the Bayes optimal classifier on the target domain.

## 2   RELATED WORKS

**Problem setting**. Online Test-Time Adaptation (OTTA), a practical learning process to deal with domain shift (Ben-David et al., 2010; Saenko et al., 2010; Lu et al., 2020), attempts to update parameters of the predictive model already trained on a source domain dataset by processing unlabeled mini-batch datasets from a target domain in a streaming manner with no access to the source domain dataset. Recently, various approaches have been proposed to contribute to OTTA.

**Adaptation methods**. Batch-normalization-based approaches (Wang et al., 2020; Gong et al., 2022; Mirza et al., 2022; Zhao et al., 2023) adjust the statistics of the batch normalization layer to update

the model's domain knowledge upon the arrival of a test mini-batch. For example, Wang et al. (2020) suggest updating the batch normalization statistics in the pre-trained model by using the estimated statistics from the online test batch. Mirza et al. (2022) stabilize the running mean and variance in batch normalization by augmenting the incoming instance to form a tiny batch and introducing the decaying momentum for the mean and variance. Gong et al. (2022) and Zhao et al. (2023) further address class bias through sampling and weighting techniques during estimating normalization statistics, respectively.

Entropy-minimization-based approaches (Zhang et al., 2022; Jing et al., 2022; Niu et al., 2023; Lee et al., 2024) conduct entropy minimization to learn domain knowledge, since a well-adapted model outputs more confident predictions on test instances. Zhang et al. (2022) focus on single-instance robustness and suggest minimizing the entropy calculated from the average output distribution of the model across various augmentations. Jing et al. (2022) utilize the entropy loss as the likelihood function and put forward a variational model perturbation approach. Moreover. Niu et al. (2023) and Lee et al. (2024) select part of the arriving instances to perform reliable entropy minimization.

Pseudo-labeling-based approaches (Iwasawa & Matsuo, 2021; Goyal et al., 2022; Shin et al., 2022; Yang et al., 2022; Döbler et al., 2023; Jang et al., 2023; Wang et al., 2023; Sun et al., 2024) attempt to generate high-quality pseudo-labels for unlabeled test instances to perform empirical risk minimization, which allows the model to absorb domain knowledge in a supervised learning manner. For instance, based on the distances in feature space. Iwasawa & Matsuo (2021) build a pseudo-prototype for each class, which has the ability to classify new samples. Goyal et al. (2022) utilize a derived conjugate pseudo label to train the model in a self-training manner. Shin et al. (2022) combine predictions from multiple modalities to generate pseudo-labels with a selective fusion strategy. Meanwhile. Yang et al. (2022) average the predictions of neighboring samples stored in a memory bank to produce soft pseudo-labels. Jang et al. (2023) intend to ensure prediction consistency between prototype-based and neighbor-based classifiers. Wang et al. (2023) aim at feature alignment and uniformity through the test-time self-distillation and memorized spatial local clustering. Sun et al. (2024) refine the generation process of pseudo-labels by integrating the previous prototype-based and nearest-neighbor methods as a prototype-based graph model.

More recently, a complementary line of work Wang et al. (2022); Döbler et al. (2023); Sójka et al. (2023), especially the effective approach RMT Döbler et al. (2023), focuses on the setting of Continual Test-Time Adaptation (CTTA) and adopts teacher-student consistency, which provides highly stable adaptation in non-stationary environments through teacher–student consistency.

**Connection to Curriculum learning**. Most previous OTTA methods only absorb domain knowledge at a coarse-grained batch level, limiting the absorption of the domain knowledge. Motivated by curriculum learning (Bengio et al., 2009; Kumar et al., 2010; Zhou et al., 2020b; Abbe et al., 2023), where a model is trained from easier instances to harder ones by emulating meaningful learning sequence in human curricula, we propose the CUPLOT framework. Compared to previous curriculum learning work (Zhou et al., 2020a; Zhang et al., 2021; Karim et al., 2023) in related fields, which primarily focus on the absorption of in-class knowledge, our proposed framework emphasizes on the systematic acquisition of domain knowledge under shift and is directly grounded in optimization dynamics and supported by solid theoretical foundations. Specifically, the adapted model is expected to first learn the easier instances containing domain knowledge relevant to the already learned domain knowledge, and then attempt to conquer the harder instances containing deep domain knowledge.

## 3 PROPOSED METHOD

### 3.1 PRELIMINARIES

**Online Test-time Adaptation**. Following the literature Yu Sun (2020); Boudiaf et al. (2022); Zhao et al. (2023); Shuaicheng Niu (2024), we consider the multi-class classification Ian Goodfellow & Bengio (2016) as the original training task for TTA. Let $\mathcal{X} \subset \mathbb{R}^q$ denote the $q$-dimensional instance space, $\mathcal{Y} = \{1, 2, \ldots, c\}$ be the label space, where $c$ is the number of classes, $\mathcal{D}_{\mathcal{S}} = \{(\boldsymbol{x}_i, y_i)\}_{i=1}^{n^0}$ be the dataset from the source domain $\mathcal{S}$, where the instance $\boldsymbol{x}_i \in \mathcal{X}$ and correct label $y_i \in \mathcal{Y}$ is independently sampled from a joint distribution $p_{\mathcal{S}}(\boldsymbol{x}, y)$, $\mathcal{D}_{\mathcal{T}} = (\mathcal{D}_{\mathcal{T}}^1, \mathcal{D}_{\mathcal{T}}^2, \ldots, \mathcal{D}_{\mathcal{T}}^T)$ be a sequence of unlabeled mini-batches from the target domain $\mathcal{T}$, where $\mathcal{D}_{\mathcal{T}}^t = \{\boldsymbol{x}_i^t\}_{i=1}^{n^t}$ is the received mini-batch dataset at the $t$-th step during test-time inference and the observed instance $\boldsymbol{x}_i^t \in \mathcal{X}$ with

unobserved correct label $y_i^t \in \mathcal{Y}$ is subject to a misaligned joint distribution $p_{\mathcal{T}}(\boldsymbol{x}, y) \neq p_{\mathcal{S}}(\boldsymbol{x}, y)$, $f(\cdot; \boldsymbol{\Theta}) : \mathcal{X} \mapsto \triangle^{c-1}$ denote the predictive model $f$ parameterized by $\boldsymbol{\Theta}$, where $\triangle^{c-1}$ is the $c$-dimensional probability simplex.

In TTA, we have completed the training of the prediction model $f$ on the source domain dataset $\mathcal{D}_{\mathcal{S}}$, and its parameters have been updated to $\boldsymbol{\Theta}^0$. Given the received mini-batch dataset $\mathcal{D}_{\mathcal{T}}^t$ at $t$-th step, we aims to update the parameters of the predictive model from $\boldsymbol{\Theta}^{t-1}$ to $\boldsymbol{\Theta}^t$, such that it could assign each instance $\boldsymbol{x}_i^t$ with its correct label $y_i^t$. Overall, TTA attempts to maximize the following objective: $\mathcal{O} = \frac{\sum_{t=1}^T \sum_{i=1}^{n^t} \mathbb{I}[y_i^t = \arg\max_{j \in \mathcal{Y}} f_j(\boldsymbol{x}_i^t; \boldsymbol{\Theta}^t)]}{\sum_{t=1}^T n^t}$, where $\mathbb{I}[\cdot]$ is the indicator function. Note that we follow the same protocol as (Yu Sun, 2020) where optimization is performed ahead of evaluation.

**Curriculum learning**. Curriculum learning improves optimization by presenting training samples in a deliberately ordered sequence. A curriculum is a permutation $\pi : \{1, \ldots, n\} \rightarrow \{1, \ldots, n\}$ that determines the learning order $(\boldsymbol{x}_{\pi(1)}, \ldots, \boldsymbol{x}_{\pi(n)})$. Denote by $\boldsymbol{\Theta}_t(\pi)$ the model parameters after sequentially observing the first $t$ examples under $\pi$. The goal is to identify an optimal ordering $\pi^\star$ that yields the generalization error after completing training. Intuitively, an effective curriculum first leverages easy or reliable instances, guiding the optimizer toward a stable region of the parameter space; subsequent harder or noisier samples are then leveraged to refine the decision boundary.

### 3.2 THE CUPLOT FRAMEWORK

Our CUPLOT framework aims to allow the predictive model to learn at the $t$-th batch with a more optimal instance sequence, thereby enabling the model to absorb the domain knowledge of the target domain more effectively. Specifically, the optimization step for the $t$-th batch is further decomposed into $K^t \in \{1, 2, \ldots, n^t\}$ ordered curricula, each of which uses only a selected subset of the received batch $\mathcal{D}_{\mathcal{T}}^t$ for the optimization of the predictive model.

Let $\mathbf{M}^t = [\boldsymbol{m}^{t,1}; \boldsymbol{m}^{t,2}; \ldots; \boldsymbol{m}^{t,K^t}]^\top \in \{0,1\}^{K^t \times n^t}$ to represent the sequence matrix of curriculum content, where the vector $\boldsymbol{m}^{t,k} = [m_1^{t,k}, m_2^{t,k}, \ldots, m_{n^t}^{t,k}] \in \{0,1\}^{n^t}$ indicates whether the instance $\boldsymbol{x}_i^t \in \mathcal{D}_{\mathcal{T}}^t$ should be included as the content of the $k$-th curriculum and participate in the $k$-th sub-step optimization. Then, the overall optimization of $\boldsymbol{\Theta}^{t-1}$ at the $t$-th batch is formulated as:

$$\boldsymbol{\Theta}^t = \boldsymbol{\Theta}^{t-1} - \alpha \sum_{i=1}^{n^t} \sum_{k=1}^{K^t} m_i^{t,k} \frac{\partial \ell(f(\boldsymbol{x}_i^t; \boldsymbol{\Theta}^{t-1,k-1}), \boldsymbol{d}_i^{t,k})}{\partial \boldsymbol{\Theta}^{t-1,k-1}}, \tag{1}$$

with the curriculum content matrix $\mathbf{M}^t$ is subject to

$$\forall 1 \leq k' \leq K^t, \sum_{k=1}^{k'} \sum_{i=1}^{n^t} m_i^{t,k} \leq n^t, \prod_{k=1}^{k'} \boldsymbol{m}^{t,k} = \boldsymbol{0}. \tag{2}$$

Here, $\alpha$ is the step size of the optimization, $\ell$ is the cross-entropy loss, and $\boldsymbol{d}_i^{t,k} = [d_{i,1}^{t,k}, d_{i,2}^{t,k}, \ldots, d_{i,c}^{t,k}] \in \mathbb{R}^c$ denotes the curriculum pseudo-label of the instance $\boldsymbol{x}_i^t$ with $\sum_{j=1}^c d_{i,j}^t = 1$. Besides, at the $k$-th curriculum within the $t$-th batch, the model parameters is updated from $\boldsymbol{\Theta}^{t-1,k-1}$ to $\boldsymbol{\Theta}^{t-1,k}$ in Eq. (1) as follows:

$$\boldsymbol{\Theta}^{t-1,k} = \boldsymbol{\Theta}^{t-1,k-1} - \alpha \sum_{i=1}^{n^t} m_i^{t,k} \frac{\partial \ell(f(\boldsymbol{x}_i^t; \boldsymbol{\Theta}^{t-1,k-1}), \boldsymbol{d}_i^{t,k})}{\partial \boldsymbol{\Theta}^{t-1,k-1}}, \tag{3}$$

where $\boldsymbol{\Theta}^{t-1,0} = \boldsymbol{\Theta}^{t-1}$ at the beginning step when $k = 1$.

Next, we program the sequence of course content $\mathbf{M}^t$ in Eq. (1) to activate our curriculum framework by resorting to gradient consistency $\boldsymbol{\mu}^{t,k} = [\mu_1^{t,k}, \mu_2^{t,k}, \ldots, \mu_{n^t}^{t,k}] \in \mathbb{R}^{n^t}$ to decide the $k$-th curriculum content $\boldsymbol{m}^{t,k}$. Adopting reverse thinking, if the gradient on an instance is more inconsistent with the overall gradient on the batch, its knowledge will be diluted and less knowledge will be absorbed by the model at the batch level. Therefore, during our more fine-grained instance-level learning, such an instance should be scheduled for later in the learning curriculum. This is in the hope that after the model has learned more domain knowledge, it will be able to effectively learn from such an instance.

On one hand, the gradient $\boldsymbol{g}_i^{t,k}$ on the instance $\boldsymbol{x}_i^t$ is calculated as follows:

$$\boldsymbol{g}_i^{t,k} = \frac{\partial \ell(f(\boldsymbol{x}_i^t; \boldsymbol{\Theta}^{t-1,k-1}), \boldsymbol{d}_i^{t,k-1})}{\partial \boldsymbol{\Theta}^{t-1,k-1}}. \tag{4}$$

On the other hand, the gradient on the content to be learned $\boldsymbol{G}^{t,k}$ is calculated as follows:

$$\boldsymbol{G}^{t,k} = \sum_{i=1}^{n^t} (\mathbf{1} - \boldsymbol{s}_i^{t,k-1}) \frac{\partial \ell(f(\boldsymbol{x}_i^t; \boldsymbol{\Theta}^{t-1,k-1}), \boldsymbol{d}_i^{t,k-1})}{\partial \boldsymbol{\Theta}^{t-1,k-1}}, \tag{5}$$

where the vector $\boldsymbol{s}^{t,k-1} = [s_1^{t,k-1}, s_2^{t,k-1}, \ldots, s_{n^t}^{t,k-1}] \in \{0,1\}^{n^t}$ denotes the cumulative curriculum content consisting of the learned instances before the $k$-th step within $t$-th batch, i.e., $\boldsymbol{s}^{t,k-1} = \sum_{k'=1}^{k-1} \boldsymbol{m}^{t,k'}$ if $k-1 \geq 1$, and thus $\mathbf{1} - \boldsymbol{s}_i^{t,k-1}$ denotes the content to be learned. When $k = 1$, we set $\boldsymbol{s}^{t,k-1} = \mathbf{0}$ and $\boldsymbol{d}_i^{t,k-1} = f(\boldsymbol{x}_i^t; \boldsymbol{\Theta}^{t-1,k-1})$.

Based on Eq. (4) and (5), the gradient consistency $\mu_i^{t,k}$ for the instance $\boldsymbol{x}_i^t$ is measured as follows:

$$\mu_i^{t,k} = \frac{1}{||\boldsymbol{G}^{t,k} - \boldsymbol{g}_i^{t,k}||_1}, \tag{6}$$

whose larger value indicates that the gradients are more consistent. In practice, when $k-1$ curriculum steps have selected samples, the remaining samples in the $k$-th step are simply all the unselected samples, and no further selection is needed.

After obtaining the gradient consistency $\boldsymbol{\mu}^{t,k}$, we generate the $k$-th curriculum content:

$$\boldsymbol{m}^{t,k} = (\mathbf{1} - \boldsymbol{s}^{t,k-1}) \cdot \psi(\boldsymbol{\mu}^{t,k}), \tag{7}$$

where $\psi : \mathbb{R}^{n^t} \mapsto \{0,1\}^{n^t}$ with $\psi_i(\boldsymbol{\mu}^{t,k}) = \mathbb{I}[\mu_i^{t,k} \geq \delta]$, and $\delta$ is a threshold employed to sieve the instance according to the gradient consistency $\boldsymbol{\mu}^{t,k}$. Practically, by considering efficiency while adapting, the threshold $\delta$ is usually set as the top-$B$ value of the vector $\boldsymbol{\mu}^{t,k} \cdot (\mathbf{1} - \boldsymbol{s}^{t,k-1})$ with $B = \text{round}(\log(\mathbf{1} - \boldsymbol{s}^{t,k-1}))$, and the number of scheduled curricula $K^t$ is set around $\log n^t$. $B$ is chosen so that after $K^t$ curriculum steps, all samples in the batch are covered, ensuring a complete and well-structured curriculum.

Then, we consider the generation of the pseudo label $\boldsymbol{d}_i^{t,k}$ in Eq. (1) and (3). The pseudo-label $\boldsymbol{d}_i^{t,k}$ deeply depends on the previously learned curriculum content, and thus is called curriculum pseudo-labeling in our framework. Specifically, if $m_i^{t,k} = 1$, the curriculum pseudo-label $\boldsymbol{d}_i^t$ of the instance $\boldsymbol{x}_i^t$ will be generated as follows:

$$\boldsymbol{d}_i^{t,k} = \text{Softmax}(\boldsymbol{z}_i^{t,k} \mathbf{W}^{t,k\top} / \tau_i^t), \tag{8}$$

where $\tau_i^t$ is introduced to control the smoothness of the curriculum pseudo-label $\boldsymbol{d}_i^t$ of the instance $\boldsymbol{x}_i^t$, $\boldsymbol{z}_i^{t,k} \in \mathbb{R}^{1 \times r}$ is a extracted feature vector in the $r$-dimensional space, and $\mathbf{W}^{t,k} = [\boldsymbol{w}_1^{t,k}, \boldsymbol{w}_2^{t,k}, \ldots, \boldsymbol{w}_c^{t,k}]^\top \in \mathbb{R}^{c \times r}$ is the $c$ class prototypes at the $k$-th curriculum. In our CU-PLOT framework, we employ a $Q$-layer neural network with the Softmax operation as the instantiation of the predictive model $f(\cdot; \boldsymbol{\Theta}) = \text{Softmax}(h(\phi(\cdot; \boldsymbol{\Theta}_{1:Q-1}); \boldsymbol{\Theta}_Q))$, where $\boldsymbol{\Theta}_{1:Q-1} = \{\boldsymbol{\Theta}_1, \boldsymbol{\Theta}_2, \ldots, \boldsymbol{\Theta}_{Q-1}\}$ denotes the parameters of the feature extractor $\phi$, $\boldsymbol{\Theta}_Q$ denotes the parameters of the last linear layer $h$. Hence, the extracted feature $\boldsymbol{z}_i^{t,k}$ is calculated by:

$$\boldsymbol{z}_i^{t,k} = \frac{\phi(\boldsymbol{x}_i^t; \boldsymbol{\Theta}_{1:Q-1}^{t-1,k-1})}{||\phi(\boldsymbol{x}_i^t; \boldsymbol{\Theta}_{1:Q-1}^{t-1,k-1})||_1}, \tag{9}$$

where the L1-norm is employed to perform normalization.

The $j$-th class prototype $\boldsymbol{w}_j^{t,k}$ in $\mathbf{W}^{t,k}$ will be calculated from the extracted features of the selected instances in the previous curricula:

$$\boldsymbol{w}_j^{t,k} = \frac{\sum_{i=1}^{n^t} \mathbb{I}[\hat{y}_i^t = j] s_i^{t,k} \boldsymbol{z}_i^{t,k}}{\sum_{i=1}^{n^t} \mathbb{I}[\hat{y}_i^t = j] s_i^{t,k}}, \tag{10}$$

---

**Algorithm 1** The CUPLOT Framework

---

**Input:** The pre-trained predictive model $f(\cdot; \boldsymbol{\Theta}^0)$, a sequence of unlabeled mini-batches $\mathcal{D}_{\mathcal{T}}$;
 1: **for** $t = 1, 2, \ldots, T$ **do**
 2:     **for** $k = 1, 2, \ldots, K^t$ **do**
 3:         Evaluate the content to be learned through gradient consistency $\boldsymbol{\mu}^{t,k}$ based on Eq. (6);
 4:         Arrange instances into the curriculum content $\boldsymbol{m}^{t,k}$ according to Eq. (7);
 5:         Generate the curriculum pseudo-label $\boldsymbol{d}_i^{t,k}$ for each instance based on Eq. (8);
 6:         Optimize the parameters of the model from $\boldsymbol{\Theta}^{t-1,k-1}$ to $\boldsymbol{\Theta}^{t-1,k}$ based on Eq. (3);
 7:     **end for**
 8: **end for**
**Output:** The predictive model $f(\cdot; \boldsymbol{\Theta}^T)$.

---

where $\hat{y}_i^t = \arg\max_{j \in \mathcal{Y}} f_j(\boldsymbol{x}_i^t; \boldsymbol{\Theta}^{t-1,k-1})$ is the prediction of the model on the instance $\boldsymbol{x}_i^t$. Practically, we follow (Wang et al., 2023) to maintain a memory bank to store the pairs of extracted features and outputs of the model, and follow (Iwasawa & Matsuo, 2021) to filter pairs which may be incorrect.

According to Eq. (8), (9), and (10), we build a strong relationship between the pseudo label $\boldsymbol{d}$ and the sequence matrix of curriculum $\mathbf{M}^t$, enabling domain knowledge to be absorbed in a more fine-grained manner. The quality of the generated curriculum pseudo-labels improves accordingly, thereby adapting the model to the test domain more effectively. The detailed algorithmic description of CUPLOT is presented in Algorithm 1.

### 3.3 THEORETICAL ANALYSIS

To demonstrate the superiority of the curriculum framework in OTTA, we first need to define a crucial concept helping us quantify the model's proximity to the Bayes optimal classifier on target domain.

**Definition 1.** *($e^\star$-adaptation ability). Let $L(\boldsymbol{\Theta}) := \{\boldsymbol{x} | y = \arg\max_{j \in \mathcal{Y}} f_j(\boldsymbol{x}; \boldsymbol{\Theta})\}$ denote instances predicted correctly by the model $f$ with the parameters $\boldsymbol{\Theta}$, and $L(e) := \{\boldsymbol{x} | p(y|\boldsymbol{x}) - p(o|\boldsymbol{x}) \leq e\}$, where $o = \arg\max_{j \in \mathcal{Y}, j \neq y} p(j|\boldsymbol{x})$, denote instance whose posterior margin between the highest and second-highest is less than $e$. We say that the model $f(\cdot; \boldsymbol{\Theta})$ has the $e$-adaptation ability on the target domain $\mathcal{T}$, if $e^\star = \arg\max_e |L(\boldsymbol{\Theta}) \cap L(e)|$, where $|\cdot|$ denotes the cardinality of a set.*

The value of $e$ can reflect the bound of the model's approaching the Bayes optimal classifier, provided that Tsybakov condition (Chaudhuri & Dasgupta, 2014; Belkin et al., 2018; Qiao et al., 2019), which quantifies how well classes are separated on the decision boundary $\{\boldsymbol{x} : p(y|\boldsymbol{x}) = p(o|\boldsymbol{x})\}$, is satisfied. Specifically, there exisits constants $C, \lambda > 0$, and $\epsilon_0 \in (0, 1)$, such that for all $\epsilon \leq \epsilon_0$, $\mathbb{P}[p(y|\boldsymbol{x}) - p(o|\boldsymbol{x}) \leq e] \leq C\epsilon^\lambda$. Then the chance of the model $f(\cdot; \boldsymbol{\Theta})$ with $e^\star$-adaptation ability to be consistent with the Bayes optimal classifier on the target domain is bounded as follows:

$$\mathbb{P}[\boldsymbol{x} \in L(\boldsymbol{\Theta})] \geq 1 - Ce^{\star\lambda}, \tag{11}$$

where we employ $O(e^\star)$ to denote the above bound.

Next, we establish the relationship between the gradient update and the proportion of correct pseudo-labels. Let $\boldsymbol{\Theta}^\star$ denote the parameters of a well-adapted classifier under the target domain distribution $p_{\mathcal{T}}(\boldsymbol{x}, y)$, $\mathcal{I} = \{i | \arg\max_{j \in \mathcal{Y}} d_{i,j} = y_i\}$ denote some instances with correct pseudo-labels, $\bar{\mathcal{I}} = \{i | \arg\max_{j \in \mathcal{Y}} d_{i,j} \neq y_i\}$ denote some instances with incorrect pseudo-labels. We make the following assumption:

**Assumption 1.** *Let $\nabla\boldsymbol{\Theta}(\mathcal{D}) = \sum_{i \in \mathcal{D}} \alpha \frac{\partial \ell(f(\boldsymbol{x}; \boldsymbol{\Theta}), \boldsymbol{d}_i)}{\partial \boldsymbol{\Theta}}$ denote the gradient of the model $f(\cdot; \boldsymbol{\Theta})$ using pseudo-labels on the instances with any index set $\mathcal{D}$. Then there exists the constant $\zeta > 0$, we have*

$$||\boldsymbol{\Theta} - \nabla\boldsymbol{\Theta}(\mathcal{D}) - \boldsymbol{\Theta}^\star|| \leq \zeta \frac{|\bar{\mathcal{I}} \cap \mathcal{D}|}{|\mathcal{I} \cap \mathcal{D}|}.$$

Assumption 1 implies that if $\boldsymbol{\Theta}$ is updated using the instances with more correct pseudo-labels in a batch, it will get closer to $\boldsymbol{\Theta}^\star$, the parameters of the Bayes optimal classifier. In contrast, if $\boldsymbol{\Theta}$ is updated using the instances with more incorrect pseudo-labels in a batch, it will move further away from $\boldsymbol{\Theta}^\star$. Then under Assumption 1, we could obtain the following theorem about the bound $O(e^\star)$:

Table 1: Classification accuracy (%) and Wilcoxon signed-ranks test results (**win**[$p$-value]) of comparing approaches on image corruption benchmarks. Full results are in Tables 17–19.

| Methods | CIFAR-10-C | CIFAR-100-C | ImageNet-C |
|---|---|---|---|
| ERM | 55.51 **win**[6.1e-5] | 34.20 **win**[6.1e-5] | 40.05 **win**[6.1e-5] |
| BN | 85.48 **win**[6.5e-4] | 56.68 **win**[6.1e-5] | – |
| TENT | 85.81 **win**[3.1e-4] | 57.21 **win**[6.1e-5] | – |
| PL | 85.91 **win**[6.5e-4] | 58.44 **win**[6.1e-5] | 49.99 **win**[6.1e-5] |
| SHOT-IM | 86.33 **win**[6.1e-5] | 59.14 **win**[1.8e-4] | 54.43 **win**[6.5e-4] |
| T3A | 59.56 **win**[6.1e-5] | $\overline{34.89}$ **win**[6.1e-5] | $\overline{39.67}$ **win**[6.1e-5] |
| TAST | 85.30 **win**[6.1e-5] | 51.52 **win**[6.1e-5] | 34.67 **win**[6.1e-5] |
| TAST-BN | 86.11 **win**[2.2e-3] | 50.92 **win**[6.1e-5] | – |
| TSD | 86.51 **win**[1.5e-3] | 58.49 **win**[8.0e-4] | 44.05 **win**[6.1e-5] |
| PROGRAM | $\overline{82.10}$ **win**[6.1e-5] | 55.63 **win**[6.1e-5] | 34.45 **win**[6.1e-5] |
| DEYO | 86.14 **win**[6.1e-5] | 59.08 **win**[6.5e-4] | 50.43 **win**[6.1e-5] |
| CUPLOT | **87.35** | **60.11** | **55.21** |

**Theorem 1.** *Suppose that the difference between $f_j(\boldsymbol{x}; \boldsymbol{\Theta})$ and $p(j|\boldsymbol{x})$ and the incorrectness of pseudo-labels is bounded by the distance between $\boldsymbol{\Theta}$ and $\boldsymbol{\Theta}^\star$, i.e., there exist the constants $\beta, \gamma > 0$, $|f_j(\boldsymbol{x}; \boldsymbol{\Theta}) - p(j|\boldsymbol{x})| \leq \beta||\boldsymbol{\Theta} - \boldsymbol{\Theta}^\star||$ and $\frac{|\overline{\mathcal{I}} \cap \mathcal{D}|}{|\mathcal{I} \cap \mathcal{D}|} \leq \gamma||\boldsymbol{\Theta} - \boldsymbol{\Theta}^\star||$. Consider an arriving batch $\mathcal{D}_\mathcal{T}^t$, the model trained with the pseudo-labels generated at the batch level has $e^\star$-adaptation ability while another model trained with the pseudo-labels derived from the curriculum framework has $e^{\star\prime}$-adaptation ability. Then, under Assumption 1, we could obtain:*

$$O(e^{\star\prime}) \geq O(e^\star). \tag{12}$$

The proof of Theorem 1 is provided in Appendix A.1. Theorem 1 shows that the chance of the model trained with our curriculum pseudo-labels to be consistent with the Bayes optimal classifier on the target domain could be bounded by a larger lower bound than that of the model trained with coarse-grained batch-level pseudo-labels. In the proof of Theorem 1, we show that the pseudo-label error $e^{\star\prime}$ obtained under our curriculum-based framework satisfies $e^{\star\prime} \leq e^\star$, where $e^\star$ corresponds to the batch-level baseline. This inequality rigorously demonstrates that our instance-level curriculum reduces the effective pseudo-label error, and thus the model under our framework converges to the Bayes-optimal classifier at a faster rate than the batch-level method.

# 4 EXPERIMENTS

## 4.1 DATASETS

Following recent advancements in online test-time adaptation (Jang et al., 2023; Sun et al., 2024), we evaluate our proposed method using a combination of image corruption benchmark datasets and domain generalization datasets. Specifically, we employ two widely employed image corruption benchmarks `CIFAR-10-C` and `CIFAR-100-C` (Hendrycks & Dietterich, 2019), and one more complex dataset `ImageNet-C`. These datasets introduce 15 types of common corruptions such as Gaussian noise and motion blurring, which are systematically applied to the test sets of `CIFAR-10`, `CIFAR-100` and `ImageNet-C` to evaluate model robustness. For training, we use the original training sets of `CIFAR-10`, `CIFAR-100` and `ImageNet` as source domains, while the highest severity level of corruption in `CIFAR-10-C` and `CIFAR-100-C` serves as the target domain. 20% of the source domain data is reserved for validation purposes.

Beside, we conduct experiments on four domain generalization benchmarks: `PACS` (Li et al., 2017) with 9991 samples and 7 classes collected from 4 domains, `VLCS` (Torralba & Efros, 2011) with 10729 samples and 5 classes collected from 4 domains, `OfficeHome` (Venkateswara et al., 2017) with 15588 samples and 65 classes collected from 4 domains, and `DomainNet` (Peng et al., 2019) with 586575 samples and 345 classes collected from 6 domains. We designate one domain as the target and treat the remaining domains as source domains. The validation set follows the same partitioning strategy as in the image corruption benchmark datasets.

## 4.2 BASELINES

We compare the performance of CUPLOT with eleven baselines frequently used for comparison in online TTA: 1) ERM (Vapnik, 1998): A baseline that directly uses the predictions of the pre-trained model on target testing instances without any adaptation. 2) BN (Schneider et al., 2020): A batch-normalization-based approach that replaces the activation statistics computed from source training instances in batch normalization layers with those computed from target testing instances. 3) TENT (Wang et al., 2020): An entropy-minimization-based approach that adapts BN layers by reducing the entropy of model predictions on target domain data. 4) PL (Lee et al., 2013): A pseudo-labeling-based approach that fine-tunes a predictive model by leveraging pseudo-labels inferred from the predictions of the model on target testing instance. 5) SHOT-IM (Liang et al., 2020): A pseudo-labeling-based approach that adapts the source encoding module by maximizing mutual information between intermediate features and classifier outputs. 6) T3A (Iwasawa & Matsuo, 2021): A pseudo-labeling-based approach that generates pseudo labels for target testing instances based on their distances to the estimated class prototypes. 7) TAST (Jang et al., 2023): A pseudo-labeling-based approach that adapts the model by aligning pseudo-labels inferred from the nearest neighbors with those inferred from class prototypes. 8) TAST-BN (Jang et al., 2023): A variation of TAST that adjusts the BN layers to adapt the model instead of updating the adaptation modules. 9) TSD (Wang et al., 2023): A pseudo-labeling-based approach that leverages a memory bank to calculate the pseudo-prototypes for every class and generate pseudo-labels for model refinement. 10) PROGRAM (Sun et al., 2024): A pseudo-labeling-based approach that connects prototypes and test samples in a graph, facilitating effective message passing among them to generate pseudo-labels. 11) DEYO (Lee et al., 2024): An entropy-minimization-based approach that enhances the model by further considering the influence of the object shape on prediction with a newly proposed confidence metric.

The backbone model of each compared method we employ is the same as previous studies (Jang et al., 2023; Sun et al., 2024) on the image corruption benchmark datasets CIFAR-10-C, CIFAR-100-C and domain generalization benchmark datasets. On the image corruption benchmark datasets CIFAR-10-C and CIFAR-100-C, we adopt ResNet-50 as the backbone model. On domain generalization benchmark datasets, we conduct evaluations using ResNet-18 and ResNet-50 architectures (He et al., 2016), both of which are equipped with batch normalization layers (Ioffe, 2015). For ImageNet-C, the ViT-B32 model is used for compared approaches. Since the ViT-B32 model is not equipped with batch normalization layers, we do not report the results on the Batch-normalization-based approaches such as BN, TENT and TAST-BN.

As for source training, on domain generalization benchmarks, the models are initialized using pre-trained parameters from ImageNet-1K (Russakovsky et al., 2015). The model is updated using the Adam optimizer with the learning rate set to $5 \times 10^{-5}$. On the image corruption benchmarks CIFAR-10-C and CIFAR-100-C, we follow (Liu et al., 2021) and pre-train ResNet-50 for 1000 epochs using a combination of the classification task with the standard cross-entropy loss and the instance discrimination task with a self-supervised loss using the SGD optimizer. To balance the two tasks, the weight for the instance discrimination task is set to 0.1. On ImageNet-C, the pre-trained parameters of the ViT-B32 model is provided by the publicly available timm library, which is pretrained on ImageNet-1K.

As for target adapting, the Adam optimizer is employed to update the model parameters, the batch size is set to 128, and the learning rate is selected from the range between $10^{-3}$ and $10^{-6}$. All hyper-parameters for the TTA setting are finalized prior to accessing any test samples. The hyper-parameters for each compared algorithm are selected according to their performance on the previously split validation datasets (Gulrajani & Lopez-Paz, 2021; Wang et al., 2023). Besides, in order to ensure the reliability of our experimental results, we conduct 3 trials with different random seeds for each compared algorithm to calculate mean and standard on domain generalization benchmarks.

## 4.3 EXPERIMENTAL RESULTS

Tables 1, 2 and 3 comprehensively present a summary of the classification accuracy achieved by each compared approach within the target domains of the benchmark datasets. Note that we do not report the results of Batch-normalization-based approaches such as BN, TENT and TAST-BN on ImageNet-C in Table 1 since the backbone model ViT-B32 is not equipped with batch normalization layers. Also, due to space limitations, we report full results with detailed mean and

Table 2: Classification accuracy of comparing approaches on domain generalization benchmarks with ResNet-18. Due to the space limit, full results could be found on Table 20, 22, 24 and 26 in Appendix A.11.

| Methods | PACS | VLCS | OfficeHome | DomainNet |
|---|---|---|---|---|
| ERM | 80.08 | 75.23 | 62.41 | 35.74 |
| BN | 83.02 | 68.74 | 62.11 | 34.90 |
| TENT | 83.28 | 69.25 | 62.30 | 35.36 |
| PL | 85.82 | 74.60 | 62.54 | 35.28 |
| SHOT-IM | 82.70 | 70.99 | 63.62 | 35.89 |
| T3A | 82.26 | 75.93 | 63.83 | 36.29 |
| TAST | 84.60 | 70.88 | 63.53 | 35.37 |
| TAST-BN | 85.39 | 75.02 | 62.33 | 35.11 |
| TSD | 87.48 | 74.81 | 63.12 | 35.50 |
| PROGRAM | 82.50 | 72.35 | 62.88 | 35.94 |
| DEYO | 86.63 | 74.05 | 63.05 | 35.36 |
| CUPLOT | **87.87** | **76.97** | **64.55** | **37.35** |

Table 3: Classification accuracy of comparing approaches on domain generalization benchmarks with ResNet-50. Due to the space limit, full results could be found on Table 21, 23, 25 and 27 in Appendix A.11.

| Methods | PACS | VLCS | OfficeHome | DomainNet |
|---|---|---|---|---|
| ERM | 85.47 | 76.64 | 67.69 | 43.29 |
| BN | 86.09 | 68.35 | 67.18 | 41.54 |
| TENT | 86.58 | 69.08 | 67.48 | 42.42 |
| PL | 86.13 | 73.81 | 67.61 | 42.38 |
| SHOT-IM | 85.35 | 69.32 | 67.98 | 43.46 |
| T3A | 86.01 | 77.41 | 68.76 | 44.11 |
| TAST | 86.56 | 68.53 | 68.70 | 42.38 |
| TAST-BN | 89.23 | 71.63 | 68.60 | 42.49 |
| TSD | 91.03 | 73.82 | 69.11 | 42.27 |
| PROGRAM | 86.44 | 68.42 | 67.99 | 43.35 |
| DEYO | 88.34 | 70.49 | 68.25 | 42.47 |
| CUPLOT | **91.11** | **78.94** | **70.30** | **44.98** |

Table 4: Classification accuracy (mean ± std) of CUPLOT and its variant CUPLOT-NM on target domains.

| Domain | Backbone | CUPLOT | CUPLOT-NM |
|---|---|---|---|
| C | | **99.41±0.15** | 97.25±0.39 |
| L | ResNet-18 | **65.61±0.50** | 62.99±1.24 |
| S | | **71.25±2.23** | 69.09±2.49 |
| V | | **71.61±1.40** | 70.22±1.79 |
| C | | **99.18±0.59** | 95.95±2.74 |
| L | ResNet-50 | **65.96±1.85** | 62.10±2.21 |
| S | | **74.12±2.62** | 71.15±2.42 |
| V | | **76.50±0.61** | 73.89±1.38 |

standard deviation in Appendix A.11. The result that achieves the best performance is highlighted in bold, and the one ranked second is underlined. From Tables 1, 2 and 3, we could conclude: 1) CUPLOT significantly attains the optimal performance among all benchmark datasets and network architectures, surpassing every compared method. 2) CUPLOT outperforms the second-ranked methods on image corruption benchmarks, and it yields an average performance increase of $0.84\%$, $0.97\%$ and $0.78\%$ on CIFAR-10-C, CIFAR-100-C and ImageNet-C, respectively. 3) CUPLOT steadily boosts the classifier's performance on domain generalization benchmarks. Specifically, it realizes an average enhancement of $1.06\%$ on DomainNet for ResNet-18 and $1.53\%$ on VLCS for ResNet-50.

### 4.4 WALL-CLOCK TIME AND MEMORY CONSUMPTION ANALYSIS

To assess wall-clock time and memory consumption, we follow (Song et al., 2023; Cai et al., 2020) and conduct comparison experiments with baselines that use gradient computation for updates on the shot noise corruption of the CIFAR-100-C dataset, employing ResNet-50 as the feature extractor with a batch size of 128. Since we use gradient consistency to drive the learning order, the backward-passes-per-sample (BPPS) is $\frac{K^t(K^t-1)}{2}$. More details could be found in Appendix A.2. The evaluation results are presented in Table 5, which demonstrates that CUPLOT maintains comparable latency and memory consumption when achieving better performance. Furthermore, we demonstrate that CUPLOT could retain practical flexibility and trade-off between effectiveness and efficiency through its curriculum parameter $K^t$ in Appendix A.6.

### 4.5 EXTENSION TO CONTINUAL TEST-TIME ADAPTATION

To assess whether curriculum-based ordering remains effective under non-stationary environment, we extend CUPLOT to the setting of Continual Test-Time Adaptation by integrating an mean-teacher (MT) framework for fair comparison. Following the RMT Döbler et al. (2023) protocol on CIFAR-100 → CIFAR-100-C (severity 5), the results in Table 6 (Appendix A.3) confirms that curriculum-driven

Table 5: Performance Comparison on CIFAR-100-C (shot noise corruption) with ResNet-50, including average wall-clock time, accuracy and memory usage.

| Methods | Wall-Clock Time (s) | Params (MB) | Activations (MB) | Total (MB) | Accuracy (%) |
|---|---|---|---|---|---|
| TENT | 5.15 | 94.82 | 1761.61 | 5517.30 | 56.33 |
| SHOT-IM | 6.86 | 94.82 | 3517.50 | 5801.62 | 58.24 |
| DEYO | 8.21 | 94.82 | 3517.50 | 5990.18 | 58.06 |
| CUPLOT | 8.58 | 94.82 | 3517.50 | 6142.76 | 59.24 |

ordering is complementary to MT-style temporal ensembling: while EMA stabilizes predictions over time, ordering controls how fine-grained domain knowledge is absorbed, and the two mechanisms jointly yield more robust online adaptation.

### 4.6 FURTHER ANALYSIS

To verify the effectiveness of the curriculum pseudo-labels in CUPLOT, we carry out an ablation study with a variant of CUPLOT, i.e., CUPLOT-NM, where the model directly learns the batch without arranging curricula by setting the threshold $\delta = \min_i \boldsymbol{\mu}^{t,1}$ and the curriculum number $K^t = 1$. As presented in Table 4, CUPLOT surpasses CUPLOT-NM across all target domains of the PACS dataset whenever using ResNet-18 and ResNet-50. More ablation details about the selection of consistency metric could be found in Appendix A.8.

Besides, we perform sensitivity analysis to examine the impact of the temperature hyper-parameter $\tau$ in the generation of pseudo-labels in Eq. (8), and the batch size in our framework using the shot noise corruption of CIFAR-10-C dataset. $\tau$ increases from 0.3 to 10, and the batch size varies from 16 to 256. As illustrated in Figure 2(a), the performance of CUPLOT remains relatively stable across a broad range, which demonstrates highly desirable robustness to deliver reliable test-time adaptation performance. Meanwhile, Figure 2(b) presents the average accuracy of various methods across different batch sizes on shot noise corruption of CIFAR-10-C dataset. From Figure 2(b), our approach consistently outperforms the other methods under varying batch sizes, which demonstrates CUPLOT could flexibly handle streaming real-world data of various sizes.

Furthermore, our proposed framework provides a novel insight into active OTTA. Different from the previous active TTA work (Gui et al., 2024), in which human experts work at the aspect of the label, CUPLOT could bring the active query at the aspect of the instance via providing the difficulty levels of domain knowledge absorption between instances. Figure 2(c) presents the test accuracy (y-axis) of a variant of CUPLOT, i.e., CUPLOT-AT on CIFAR-10-C and CIFAR-100-C, where the different severity levels of corruption are mixed to serve as the target domain, and CUPLOT-AT has access to the difficulty levels of a certain proportion of data (x-axis). CUPLOT-AT arranges the data with lower difficulty levels in the earlier curricula for priority learning as much as possible. As illustrated in Figure 2(c), the performance of our framework could be further improved when the human experts provide information on the aspect of the instance if the difficulty levels of a larger proportion of instances are known, which is a nice property for those that require human interaction to improve the designed algorithm.

## 5 CONCLUSION

In this study, we proposed the CUPLOT, a novel online test-time adaptation framework, aiming to address the issue that most existing Online Test-Time Adaptation (OTTA) methods only exploit domain knowledge at a coarse-grained batch level. CUPLOT mines domain knowledge at a fine-grained instance level by organizing the arrived batch into a series of curricula based on the modeled relevance of domain knowledge between the model and instances, and enabling the model to learn instances in an orderly manner using pseudo-labels generated by class prototypes. Theoretically, we demonstrated that the model trained with curriculum pseudo-labels has a larger lower bound of the probability of being consistent with the Bayes optimal classifier on the target domain, indicating stronger adaptation ability. Extensive experiments varify the effectiveness of our proposed framework.

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

# A  TECHNICAL APPENDICES AND SUPPLEMENTARY MATERIAL

## A.1  PROOFS OF THEOREM 1

**Theorem 1.** *Suppose that the difference between $f_j(\boldsymbol{x}; \boldsymbol{\Theta})$ and $p(j|\boldsymbol{x})$ and the incorrectness of pseudo-labels is bounded by the distance between $\boldsymbol{\Theta}$ and $\boldsymbol{\Theta}^\star$, i.e., there exist the constants $\beta, \gamma > 0$, $|f_j(\boldsymbol{x}; \boldsymbol{\Theta}) - p(j|\boldsymbol{x})| \leq \beta ||\boldsymbol{\Theta} - \boldsymbol{\Theta}^\star||$ and $\frac{|\bar{\mathcal{I}} \cap \mathcal{D}|}{|\mathcal{I} \cap \mathcal{D}|} \leq \gamma ||\boldsymbol{\Theta} - \boldsymbol{\Theta}^\star||$. Consider an arriving batch $\mathcal{D}_\mathcal{T}^t$, the model trained with the pseudo-labels generated at the batch level has $e^\star$-adaptation ability while another model trained with the pseudo-labels derived from the curriculum framework has $e^{\star\prime}$-adaptation ability. Then, under Assumption 1, we could obtain:*

$$O(e^{\star\prime}) \geq O(e^\star).$$

*Proof.* We start by clarifying the key concepts and notations relevant to the proof. According to Definition 1, the $e^\star$-adaptation ability of the model $f(\cdot; \boldsymbol{\Theta})$ is determined by $e^\star = \arg\max_e |L(\boldsymbol{\Theta}) \cap L(e)|$, and we know that $\mathbb{P}[\boldsymbol{x} \in L(\boldsymbol{\Theta})] \geq 1 - Ce^{\star\lambda} = O(e^\star)$. To prove $O(e^{\star\prime}) \geq O(e^\star)$, we aim to show that $e^{\star\prime} \leq e^\star$ since $O(e)$ is a decreasing function of $e$. Here, $e^\star$ corresponds to the model trained with batch-level pseudo-labels, and $e^{\star\prime}$ corresponds to the model trained with curriculum-framework pseudo-labels.

First, we analyze the difference in model predictions. Given the condition $|f_j(\boldsymbol{x}; \boldsymbol{\Theta}) - p(j|\boldsymbol{x})| \leq \beta ||\boldsymbol{\Theta} - \boldsymbol{\Theta}^\star||$, consider the model trained with curriculum-framework pseudo-labels $f(\cdot; \boldsymbol{\Theta}_{curriculum})$ and the model trained with batch-level pseudo-labels $f(\cdot; \boldsymbol{\Theta}_{batch})$.

Next, we show that the curriculum framework can make better use of correct pseudo-labeled samples. Let $D^k$ be the cumulative instances in the curriculum framework. From the perspective of parameter update, since the curriculum-framework model is closer to $\boldsymbol{\Theta}^\star$ at the $k-1$-th step, according to the inequality $||\boldsymbol{\Theta} - \nabla\boldsymbol{\Theta}(\mathcal{D}^{k-1}) - \boldsymbol{\Theta}^\star|| \leq \zeta \frac{|\bar{\mathcal{I}} \cap \mathcal{D}^{k-1}|}{|\mathcal{I} \cap \mathcal{D}^{k-1}|}$ in Assumption 1, when the update is carried out at the $k$-th step, $\frac{|\bar{\mathcal{I}} \cap \mathcal{D}^k|}{|\mathcal{I} \cap \mathcal{D}^k|}$ will further get smaller and the curriculum-framework model will further narrow the distance from $\boldsymbol{\Theta}^\star$, that is, $||\boldsymbol{\Theta}_{curriculum}^k - \boldsymbol{\Theta}^\star|| \leq ||\boldsymbol{\Theta}_{curriculum}^{k-1} - \boldsymbol{\Theta}^\star||$. This implies that the curriculum-framework model can make better use of correct pseudo-labeled samples by setting reasonable curriculum number. By Assumption 1, updating with more correct pseudo-labels makes the model further approach $\boldsymbol{\Theta}^\star$. Thus, we have

$$||\boldsymbol{\Theta}_{curriculum} - \boldsymbol{\Theta}^\star|| \leq ||\boldsymbol{\Theta}_{batch} - \boldsymbol{\Theta}^\star||. \tag{13}$$

From this, we can infer that

$$|f_j(\boldsymbol{x}; \boldsymbol{\Theta}_{curriculum}) - p(j|\boldsymbol{x})| \leq |f_j(\boldsymbol{x}; \boldsymbol{\Theta}_{batch}) - p(j|\boldsymbol{x})|, \tag{14}$$

which indicates that the predictions of the model trained with the curriculum framework are closer to the true probability distribution $p(j|\boldsymbol{x})$.

We analyze the difference in the error rates of pseudo-labels. According to $\frac{|\bar{\mathcal{I}} \cap \mathcal{D}|}{|\mathcal{I} \cap \mathcal{D}|} \leq \gamma ||\boldsymbol{\Theta} - \boldsymbol{\Theta}^\star||$, because $||\boldsymbol{\Theta}_{curriculum} - \boldsymbol{\Theta}^\star|| \leq ||\boldsymbol{\Theta}_{batch} - \boldsymbol{\Theta}^\star||$, the error rate of pseudo-labels in the curriculum-framework training is lower. That is, the proportion of mislabeled samples in the total samples for the model trained with the curriculum framework is smaller.

Then, we combine the above-mentioned facts with the definition to derive the inequality. According to Definition 1, $L(\boldsymbol{\Theta}) = \{\boldsymbol{x}|y = \arg\max_{j \in \mathcal{Y}} f_j(\boldsymbol{x}; \boldsymbol{\Theta})\}$ and $L(e) = \{\boldsymbol{x}|p(y|\boldsymbol{x}) - p(o|\boldsymbol{x}) \leq e\}$. Since the model trained with the curriculum framework has more accurate predictions and a lower error rate of pseudo-labels, the number of instances that satisfy both $y = \arg\max_{j \in \mathcal{Y}} f_j(\boldsymbol{x}; \boldsymbol{\Theta}_{curriculum})$ and $p(y|\boldsymbol{x}) - p(o|\boldsymbol{x}) \leq e$ is relatively larger.

Specifically, we have

$$|L(\boldsymbol{\Theta}_{curriculum}) \cap L(e)| \geq |L(\boldsymbol{\Theta}_{batch}) \cap L(e)|. \tag{15}$$

When calculating $e^\star = \arg\max_e |L(\boldsymbol{\Theta}) \cap L(e)|$, for the model trained with the curriculum framework, $e^{\star\prime}$ makes $|L(\boldsymbol{\Theta}_{curriculum}) \cap L(e^{\star\prime})|$ reach its maximum value. And because

$$|L(\boldsymbol{\Theta}_{curriculum}) \cap L(e^{\star\prime})| \geq |L(\boldsymbol{\Theta}_{batch}) \cap L(e^\star)|, \tag{16}$$

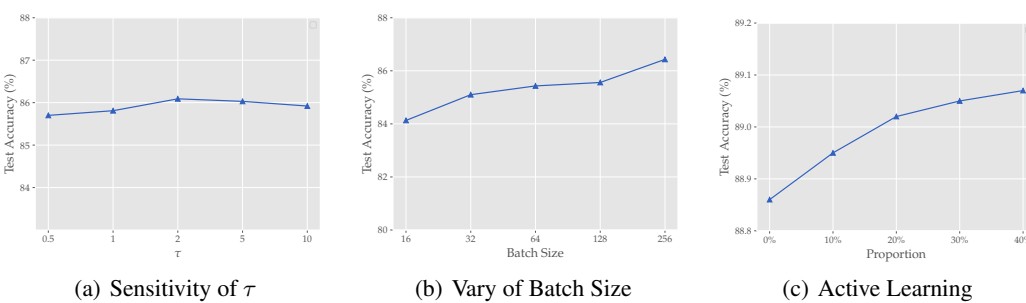

(a) Sensitivity of $\tau$       (b) Vary of Batch Size       (c) Active Learning

Figure 2: The parameter sensitivity analysis for CUPLOT.

Table 9: GPU time and classification accuracy induced by curriculum learning with varying $K^t$ on shot noise of CIFAR-10-C.

| $K^t$ | 1 | 2 | 3 | 4 |
|---|---|---|---|---|
| Time | 6.10 | 7.55 | 8.37 | 8.97 |
| Acc. | 85.04 | 86.06 | 86.21 | 86.45 |

Table 10: GPU time and classification accuracy induced by curriculum learning with varying $K^t$ on clipart subset of DomainNet.

| $K^t$ | 1 | 2 | 3 | 4 |
|---|---|---|---|---|
| Time | 102.18 | 113.32 | 126.59 | 137.38 |
| Acc. | 50.82 | 51.77 | 51.89 | 52.07 |

we can conclude that

$$e^{\star\prime} \leq e^{\star}. \tag{17}$$

Finally, since $O(e^{\star}) = 1 - Ce^{\star\lambda}$ is a monotonically decreasing function of $e$, we can obtain $O(e^{\star\prime}) \geq O(e^{\star})$. Therefore, Theorem 1 is proved.

Table 6: Classification **error rate** (%) on the CIFAR-100→CIFAR-100-C online continual test-time adaptation task under the highest corruption severity level (severity 5), evaluated on ResNeXt-29 following the Döbler et al. (2023) protocol.

| Methods | Gaussian | shot | impulse | defocus | glass | motion | zoom | snow | frost | fog | brightness | contrast | elastic | pixelate | jpeg | Mean |
|---|---|---|---|---|---|---|---|---|---|---|---|---|---|---|---|---|
| Source | 73.0 | 68.0 | 39.4 | 29.3 | 54.1 | 30.8 | 28.9 | 39.5 | 45.8 | 50.3 | 29.5 | 55.1 | 37.2 | 74.7 | 41.2 | 46.4 |
| BN | 42.1 | 40.7 | 42.7 | 27.6 | 41.9 | 29.7 | 27.9 | 34.9 | 35.0 | 41.5 | 26.5 | 30.3 | 35.7 | 32.9 | 41.2 | 35.4 |
| TENT | 37.2 | 35.8 | 41.7 | 37.9 | 51.2 | 48.3 | 45.8 | 58.4 | 63.7 | 71.1 | 70.4 | 82.3 | 88.0 | 88.5 | 90.4 | 60.9 |
| CoTTA | 40.1 | 37.7 | 39.7 | 26.9 | 38.0 | 27.9 | 26.4 | 32.8 | 31.8 | 40.3 | 24.7 | 26.9 | 32.5 | 28.3 | 33.5 | 32.5 |
| RMT | 38.5 | 34.4 | 35.4 | 26.4 | 32.7 | 27.0 | 25.0 | 27.6 | 27.5 | 30.0 | 24.0 | 25.8 | 27.0 | 25.2 | 28.4 | 29.0 |
| Ours | **38.0** | **34.0** | **34.8** | **26.0** | **32.2** | **26.4** | **24.5** | **27.2** | **27.1** | **29.5** | **23.5** | **25.3** | **26.6** | **24.7** | **27.9** | **28.5** |

## A.2 WALL-CLOCK TIME AND MEMORY CONSUMPTION ANALYSIS

We measured the wall-clock time and memory usage of our approach and baselines following EcoTTA (Song et al., 2023). Specifically, all methods are performed on a CPU constrained to 2 cores and 4 threads to emulate computationally constrained scenarios. We measured the average wall-clock time per batch using `time.perf_counter`, recording the wall-clock time before and after the execution of the TTA algorithm and averaging over all batches. The parameter and activation memory costs are measured following the TinyTL (Cai et al., 2020) codebase, and the total memory usage is tracked via `memory_profiler.memory_usage` with an interval of 0.01 seconds. The results are shown in Table 5.

## A.3 EXTENSION TO CONTINUAL TEST-TIME ADAPTATION

To examine whether curriculum-based ordering remains beneficial under strong Mean-Teacher (MT)–style stabilization, we further extend CUPLOT to Continual Test-Time Adaptation by integrating an EMA teacher into our adaptation pipeline. MT-based approaches such as COTTAWang et al. (2022) and RMTDöbler et al. (2023) stabilize online adaptation by smoothing predictions over time via teacher–student consistency. Since our curriculum mechanism is orthogonal to this

Table 7: Classification accuracy of active learning variants on CIFAR-10-M35.

| Sampling Rate | 10% | 20% | 0.3% | 0.4% |
|---|---|---|---|---|
| CUPLOT-AT | 86.62 | 86.72 | 86.83 | 86.96 |
| CUPLOT-E | 86.56 | 86.64 | 86.70 | 86.77 |
| CUPLOT-G | 86.53 | 86.62 | 86.69 | 86.75 |
| CUPLOT-M | 86.53 | 86.59 | 86.66 | 86.71 |

Table 8: Classification accuracy of our approach and compared methods on real-world temporal-shift datasets.

| Dataset | ERM | BN | TENT | PL | SHOT-IM | TSD | CUPLOT |
|---|---|---|---|---|---|---|---|
| Yearbook | 81.30 | 84.54 | 84.53 | 84.67 | 85.17 | 85.11 | 85.53 |
| EVIS | 56.59 | 45.72 | 45.73 | 45.78 | 45.93 | 46.01 | 56.87 |

stabilization—targeting the ordering of incoming samples rather than temporal ensembling—we evaluate whether ordering still brings improvements when combined with an EMA teacher.

Following the Döbler et al. (2023) evaluation protocol, we conduct experiments on the CIFAR-100→CIFAR-100-C continual test-time adaptation task at corruption severity 5 using ResNeXt-29. Table 6 reports classification error rates across all 15 corruption types. These results demonstrate that ordering benefits do not vanish under MT-style smoothing; instead, EMA-based prediction averaging and curriculum-driven sample ordering are complementary, providing cumulative improvements in adaptation stability and accuracy.

### A.4 COMPARISON TO COMMON ACTIVE LEARNING STRATEGIES

We conducted additional experiments comparing CUPLOT-AT (gradient-consistency-based instance selection) against three active learning variants equipped with different sampling criteria commonly used active learning (Huang et al., 2010; Yan et al., 2016), including:

- CUPLOT-E (entropy-based): $\text{Score}(x_i) = \sum_{j=1}^{C} d_i^j \log d_i^j$,
- CUPLOT-G (margin-based): $\text{Score} = d_i^m - d_i^o$, where $m = \arg\max_{j \in \mathcal{Y}} d_i^j$ and $o = \arg\max_{j \in \mathcal{Y}, j \neq m} d_i^j$.
- CUPLOT-M (maximum-based): $\text{Score} = d_i^m$

Similar to CUPLOT-AT, samples with higher scores are prioritized for earlier curricula, while those with lower scores are scheduled for later curricula. We manually create a dataset `CIFAR-10-M35` by mixing samples from `CIFAR-10-C` with difficulty levels 3 and 5. Table 7 illustrates the performance of these active learning variants on `CIFAR-10-M35`. From Table 7, we could observe the superiority of gradient-consistency-based instance selection.

Table 11: Classification accuracy of our approach and compared methods on shot noise of CIFAR-10-C under different batch sizes.

| Methods | 2 | 4 | 8 | 16 | 32 | 64 | 128 | 256 | 512 |
|---|---|---|---|---|---|---|---|---|---|
| SHOT-IM | 63.17 | 72.32 | 78.43 | 82.13 | 83.62 | 85.00 | 85.04 | 84.41 | 85.61 |
| DEYO | 63.77 | 72.93 | 79.08 | 82.35 | 83.86 | 84.82 | 84.58 | 84.87 | 85.09 |
| CUPLOT | 66.87 | 74.40 | 80.42 | 84.01 | 85.03 | 85.93 | 86.06 | 85.97 | 86.15 |

Table 12: Classification accuracy of different consistency metrics on CIFAR-100-C.

| Criterion | Noise | Blur | Weather | Digital |
|-----------|-------|------|---------|---------|
| Gradient Consistency | 55.16 | 64.01 | 58.80 | 62.70 |
| Uncertainty | 54.79 | 63.76 | 58.66 | 62.55 |
| Cross-entropy Loss | 54.85 | 63.69 | 58.63 | 62.61 |

Table 13: Classification accuracy of different consistency metrics on PACS.

| Criterion | A | C | P | S |
|-----------|------|------|------|------|
| Gradient Consistency | 91.33 | 90.00 | 97.60 | 85.51 |
| Uncertainty | 90.92 | 89.84 | 97.68 | 85.22 |
| Cross-entropy Loss | 90.81 | 89.95 | 97.55 | 85.15 |

## A.5 PRACTICAL APPLICABILITY OF OUR METHOD

To assess CUPLOT's real-world applicability, we conducted additional experiments on two temporal-shift datasets that reflect natural, non-synthetic distribution shifts:

- `Yearbook`: A long-span dataset of high school portraits spanning eight decades, characterized by evolving demographics, camera technologies, and visual styles.
- `EVIS`: A dataset of electronic product and vehicle images, indexed by upload dates to capture real-world trends and domain drift.

These datasets simulate realistic test-time adaptation scenarios where the target domain shifts over time and is not seen during training. As shown below in Table 8, CUPLOT significantly outperforms existing TTA methods, demonstrating its ability to generalize and adapt in complex real-life settings.

## A.6 TRADE-OFF BETWEEN EFFECTIVENESS AND EFFICIENCY

CUPLOT retains practical flexibility and trade-off between effectiveness and efficiency through its curriculum parameter $K^t$ (number of curricula per batch), empirically defined as $K^t = \text{round}(\log n^t)$ by default. Reducing $K^t$ (e.g., setting $K^t = 1$ to mimic batch-level learning) significantly lowers computational cost while retaining performance gains to some extent. This allows users to tailor $K^t$ to resource constraints, balancing efficiency and accuracy. Table 9 and Table 10 report the running time when $K^t$ varies from $[1, 4]$ on `CIFAR-10-C` using ResNet-50 and `DomainNet` using ResNet-18, demonstrating CUPLOT's practical flexibility and trade-off between effectiveness and efficiency.

## A.7 CUPLOT'S PERFORMANCE UNDER VARYING BATCH SIZES

Table 11 presents the classification accuracy of our approach and compared methods on shot noise of `CIFAR-10-C` under different batch sizes. These results demonstrate that our method achieves consistently high accuracy across a wide range of batch sizes.

## A.8 PERFORMANCE WITH OTHER METRICS

We conducted ablation studies comparing gradient consistency with entropy. Table 12 and 13 presents the accuracy of different metrics on CIFAR-100-C and PACS, respectively. From the tables, we validate the effectiveness of gradient consistency compared to entropy.

Additionally, we empirically conduct an ablation comparing gradient consistency with another two alternatives: (1) Softmax Confidence: For sample $x_i$, we compute the predicted class probabilities $q_i = \text{softmax}(f_\theta(\boldsymbol{x}_i))$. The confidence score is defined as $s_i = \max_j q_i^j$. Samples with lower confidence are considered harder and are scheduled later in the curriculum. (2) Feature-space Similarity (Prototype-based Selection): For each class $j$, we maintain a prototype vector $w_j$, typically

Table 14: Classification accuracy of different consistency metrics on CIFAR-100-C and DomainNet.

| Methods | CIFAR-100-C | DomainNet |
|---|---|---|
| Gradient Consistency | **60.11** | **37.35** |
| Confidence | 59.87 | 37.23 |
| Feature | 59.72 | 37.07 |

computed as the mean feature representation of samples in the batch. For sample $x_i$, we extract its feature $z_i$ and compute its distance to the nearest class prototype: $s_i = \min_j \|\phi_\theta(\boldsymbol{x}_i) - \boldsymbol{w}_j\|_2$. Samples with larger distances are considered harder and scheduled later in the curriculum, while samples closer to the prototype are scheduled earlier. Table 14 validates the effectiveness of gradient consistency compared to confidence-based and feature-level selection strategies.

A.9 PERFORMANCE WITH OTHER SELECTION STRATEGIES

To validate the effectiveness of our curriculum-based scheduling, we compare it with random selection and sequential (input-order) selection under the same test-time adaptation setup. Results are shown in Table 15. Both random and sequential orders achieve similar performance, whereas our gradient-consistency–driven curriculum consistently improves accuracy. This demonstrates that CUPLOT's gains are not due to arbitrary ordering but result from the proposed principled curriculum.

Table 15: Accuracy of different sequence strategies.

| Methods | CIFAR-10-C | CIFAR-100-C | ImageNet-C |
|---|---|---|---|
| Sequential Selection | 86.52 | 58.54 | 44.01 |
| Random Selection | 86.47 | 58.50 | 44.06 |
| CUPLOT | **87.35** | **60.11** | **55.21** |

A.10 PLUG-AND-PLAY STUDY

We conduct additional experiments integrating our gradient-consistency curriculum with two representative baselines: SHOT-IMLiang et al. (2020) and DEYOLee et al. (2024). In both cases, we replace their batch-level pseudo-labeling step with CUPLOT's curriculum-based ordering while keeping all other components unchanged. The results in Table 16 support that the proposed curriculum is a general and complementary mechanism that enhances other TTA methods—not tied to our prototype-based instantiation.

Table 16: Accuracy improvements when integrating CUPLOT's curriculum into SHOT-IMLiang et al. (2020) and DEYOLee et al. (2024) across corruption benchmarks.

| Methods | CIFAR-10-C | CIFAR-100-C | ImageNet-C |
|---|---|---|---|
| SHOT-IM | 86.33 | 59.14 | 54.43 |
| SHOT-IM+Ours | **86.95** | **59.87** | **55.25** |
| DeYO | 86.14 | 59.08 | 50.43 |
| DeYO+Ours | **87.01** | **59.60** | **51.53** |

A.11 FULL EXPERIMENTAL RESULTS

Table 17, 18, 19, 20, 21, 22, 23,24, 25, 26, and 27 present full results of each compared approach on datasets CIFAR-10-C, CIFAR-100-C, ImageNet-C, PACS, VLCS, OfficeHome, and DomainNet, respectively. Also, we present tSNE (Van der Maaten & Hinton, 2008) visualizations on the domain A of the benchmark dataset PACS for both the ERM baseline and our proposed framework CUPLOT, as depicted in Figure 3 in Appendix A.11. Once adapted to the target domain,

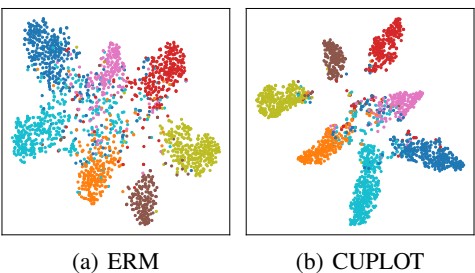

(a) ERM          (b) CUPLOT

Figure 3: tSNE visualization on PACS domain A.

Table 17: Full results on the CIFAR-10-C dataset.

| Methods | shot | motion | snow | pixelate | gaussian | defocus | brightness | fog | zoom | frost | glass | impulse | coontrast | jpeg | elastic | Avg. |
|---|---|---|---|---|---|---|---|---|---|---|---|---|---|---|---|---|
| ERM | 42.14 | 57.12 | 72.50 | 59.61 | 35.62 | 73.83 | 88.55 | 49.87 | 78.97 | 63.55 | 33.44 | 22.02 | 20.61 | 72.33 | 62.49 | 55.51 |
| BN | 84.05 | 87.30 | 85.51 | 89.40 | 82.69 | 90.99 | 92.33 | 83.31 | 92.63 | 87.99 | 75.17 | 72.84 | 89.86 | 85.87 | 82.33 | 85.48 |
| TENT | 84.38 | 87.53 | 85.76 | 89.62 | 83.12 | 91.11 | 92.44 | 83.87 | 92.77 | 88.10 | 75.83 | 73.46 | 90.53 | 86.12 | 82.52 | 85.81 |
| PL | 84.34 | 87.83 | 86.28 | 88.93 | 82.98 | 90.41 | 91.71 | 85.49 | 91.95 | 87.63 | 76.77 | 76.53 | 90.65 | 84.91 | 82.22 | 85.91 |
| SHOT-IM | 85.04 | 88.01 | 86.86 | 89.12 | 83.54 | 91.01 | 91.80 | 85.92 | 91.92 | 88.22 | 77.84 | 76.06 | 91.08 | 85.82 | 82.73 | 86.33 |
| T3A | 50.74 | 60.54 | 72.92 | 65.48 | 45.06 | 75.71 | 88.16 | 52.64 | 80.58 | 65.11 | 41.35 | 29.22 | 24.55 | 73.99 | 67.29 | 59.56 |
| TAST | 83.85 | 87.32 | 85.57 | 88.90 | 82.65 | 90.98 | 91.99 | 83.01 | 92.23 | 87.57 | 75.38 | 72.49 | 89.44 | 85.73 | 82.41 | 85.30 |
| TAST-BN | 84.87 | 87.94 | 86.10 | 89.98 | 83.62 | 91.27 | 92.45 | 84.21 | 92.76 | 88.30 | 76.50 | 74.28 | 89.83 | 86.35 | 83.14 | 86.11 |
| TSD | 85.02 | 88.30 | 86.69 | 89.94 | 83.96 | 91.40 | 92.60 | 85.12 | 92.94 | 88.61 | 76.92 | 75.05 | 91.10 | 86.79 | 83.18 | 86.51 |
| PROGRAM | 81.31 | 84.15 | 82.41 | 86.06 | 78.83 | 87.99 | 89.22 | 80.05 | 89.67 | 84.57 | 71.02 | 68.91 | 86.09 | 82.64 | 78.61 | 82.10 |
| DEYO | 84.58 | 87.78 | 87.01 | 88.68 | 83.48 | 90.35 | 91.71 | 85.94 | 92.15 | 87.48 | 76.66 | 76.38 | 91.32 | 86.05 | 82.54 | 86.14 |
| CUPLOT | 86.06 | 89.08 | 87.91 | 89.93 | 84.57 | 91.52 | 92.40 | 87.45 | 92.66 | 89.02 | 79.11 | 77.42 | 91.54 | 87.24 | 84.29 | 87.35 |

CUPLOT is capable of generating extracted features that are more clearly separated. These clearly indicate the significance of curriculum pseudo-labels in enhancing absorption of domain knowledge when the model is adapting.

## A.12 THE USE OF LARGE LANGUAGE MODELS

We acknowledge the use of a large language model (LLM) as an assistive tool during the preparation of this manuscript. The LLM's role was strictly limited to language-related refinements: specifically, it aided in grammar and spelling corrections, and helped enhance the logical coherence and readability of the prose. Additionally, the model provided support for generating certain segments of code. It is important to emphasize that the core conceptual framework, theoretical analyses, experimental design, and conclusions presented in this paper are the original work of the authors, with no involvement of the LLM in shaping these substantive research components.

Table 18: Full results on the CIFAR-100-C dataset.

| Methods | shot | motion | snow | pixelate | gaussian | defocus | brightness | fog | zoom | frost | glass | impulse | coontrast | jpeg | elastic | Avg. |
|---|---|---|---|---|---|---|---|---|---|---|---|---|---|---|---|---|
| ERM | 33.09 | 32.31 | 39.60 | 43.78 | 31.04 | 43.82 | 54.76 | 14.61 | 48.65 | 36.84 | 23.74 | 16.66 | 7.75 | 49.73 | 36.57 | 34.20 |
| BN | 55.63 | 58.14 | 54.38 | 64.50 | 55.53 | 63.92 | 65.13 | 43.27 | 67.74 | 58.67 | 48.19 | 42.50 | 56.30 | 62.56 | 53.77 | 56.68 |
| TENT | 56.33 | 58.70 | 54.62 | 64.77 | 55.92 | 64.41 | 65.61 | 44.25 | 68.03 | 59.09 | 48.86 | 43.07 | 57.35 | 62.89 | 54.30 | 57.21 |
| PL | 57.85 | 60.27 | 55.43 | 65.22 | 56.81 | 64.96 | 65.96 | 46.84 | 68.41 | 59.69 | 50.68 | 44.75 | 60.77 | 62.77 | 56.18 | 58.44 |
| SHOT-IM | 58.24 | 60.59 | 56.40 | 65.73 | 57.61 | 65.88 | 66.88 | 47.54 | 69.02 | 60.71 | 50.84 | 46.00 | 61.27 | 64.03 | 56.38 | 59.14 |
| T3A | 35.29 | 33.25 | 38.78 | 44.16 | 32.68 | 44.16 | 54.31 | 16.28 | 48.60 | 37.26 | 25.35 | 18.21 | 7.94 | 49.03 | 38.11 | 34.89 |
| TAST | 51.34 | 52.26 | 48.48 | 57.81 | 50.74 | 58.65 | 58.70 | 39.86 | 61.22 | 53.55 | 43.9 | 39.74 | 51.30 | 56.03 | 49.16 | 51.52 |
| TAST-BN | 50.42 | 52.32 | 47.83 | 57.69 | 49.93 | 57.70 | 58.07 | 39.33 | 60.71 | 52.31 | 43.48 | 38.95 | 51.03 | 55.67 | 48.38 | 50.92 |
| TSD | 57.74 | 59.81 | 55.89 | 65.54 | 57.20 | 65.65 | 66.62 | 46.21 | 68.91 | 60.14 | 49.90 | 44.51 | 60.26 | 63.57 | 55.40 | 58.49 |
| PROGRAM | 54.68 | 56.85 | 53.61 | 62.82 | 54.38 | 62.58 | 64.05 | 42.43 | 66.95 | 58.15 | 46.98 | 41.76 | 55.18 | 61.75 | 52.32 | 55.63 |
| DEYO | 58.06 | 61.13 | 56.18 | 65.30 | 57.13 | 65.75 | 66.42 | 48.41 | 68.64 | 59.77 | 51.26 | 45.87 | 62.13 | 63.52 | 56.61 | 59.08 |
| CUPLOT | 59.24 | 61.58 | 57.11 | 65.92 | 58.41 | 66.49 | 66.52 | 50.46 | 69.56 | 61.12 | 52.57 | 47.82 | 62.93 | 64.43 | 57.53 | 60.11 |

Table 19: Full results on the ImageNet-C dataset.

| Methods | shot | motion | snow | pixelate | gaussian | defocus | brightness | fog | zoom | frost | glass | impulse | coontrast | jpeg | elastic | Avg. |
|---|---|---|---|---|---|---|---|---|---|---|---|---|---|---|---|---|
| ERM | 46.10 | 36.70 | 40.66 | 61.72 | 43.90 | 27.58 | 69.36 | 31.10 | 30.24 | 41.76 | 21.58 | 44.12 | 4.62 | 59.66 | 41.62 | 40.05 |
| BN | - | - | - | - | - | - | - | - | - | - | - | - | - | - | - | - |
| TENT | - | - | - | - | - | - | - | - | - | - | - | - | - | - | - | - |
| PL | 53.78 | 50.18 | 53.52 | 69.10 | 52.64 | 42.52 | 73.08 | 47.72 | 39.76 | 50.06 | 40.66 | 51.74 | 3.68 | 66.66 | 54.80 | 49.99 |
| SHOT-IM | 55.74 | 53.60 | 55.40 | 69.76 | 54.00 | 45.00 | 73.30 | 51.66 | 48.54 | 53.72 | 48.18 | 54.00 | 27.24 | 67.82 | 58.54 | 54.43 |
| T3A | 45.94 | 36.58 | 40.72 | 61.80 | 43.84 | 27.34 | 69.36 | 28.00 | 30.22 | 41.50 | 21.12 | 44.04 | 3.26 | 59.60 | 41.70 | 39.67 |
| TAST | 38.90 | 31.32 | 36.10 | 53.44 | 37.04 | 23.46 | 61.32 | 27.90 | 26.62 | 36.44 | 17.80 | 37.18 | 3.90 | 50.56 | 38.06 | 34.67 |
| TAST-BN | - | - | - | - | - | - | - | - | - | - | - | - | - | - | - | - |
| TSD | 55.02 | 52.26 | 24.90 | 69.74 | 53.64 | 41.80 | 73.30 | 9.30 | 30.00 | 28.86 | 44.42 | 54.00 | 0.52 | 67.66 | 55.26 | 44.05 |
| PROGRAM | 46.58 | 30.14 | 26.04 | 62.60 | 43.36 | 32.40 | 67.42 | 3.94 | 34.72 | 16.06 | 3.18 | 43.64 | 1.18 | 60.40 | 45.02 | 34.45 |
| DEYO | 54.06 | 50.44 | 52.80 | 69.14 | 52.50 | 42.12 | 73.42 | 45.80 | 42.36 | 52.26 | 42.36 | 52.50 | 4.62 | 67.12 | 55.02 | 50.43 |
| CUPLOT | 56.10 | 54.40 | 56.48 | 70.34 | 54.02 | 46.44 | 73.56 | 52.78 | 50.02 | 54.76 | 49.38 | 54.34 | 28.12 | 68.08 | 59.32 | 55.21 |

Table 20: Full results on the PACS dataset with ResNet-18.

| Methods | A | C | P | S | Avg. |
|---|---|---|---|---|---|
| ERM | 78.92±1.59 | 76.42±3.24 | 94.79±0.63 | 70.20±1.40 | 80.08 |
| BN | 82.36±0.37 | 81.41±0.79 | 95.87±0.10 | 72.42±0.77 | 83.02 |
| TENT | 82.55±0.37 | 81.60±0.74 | 96.03±0.15 | 72.92±0.56 | 83.28 |
| PL | 85.63±0.82 | 84.47±0.45 | 95.89±0.54 | 77.30±1.91 | 85.82 |
| SHOT-IM | 85.19±1.02 | 81.25±1.00 | 95.91±1.02 | 68.45±1.64 | 82.70 |
| T3A | 80.71±1.48 | 79.29±2.42 | 95.93±0.52 | 73.09±1.13 | 82.26 |
| TAST | 84.31±0.52 | 82.95±0.64 | 96.75±0.21 | 74.40±0.40 | 84.60 |
| TAST-BN | 84.80±1.12 | 83.15±0.62 | 96.63±0.46 | 76.96±0.99 | 85.39 |
| TSD | 87.92±0.62 | 86.79±0.18 | 96.65±0.52 | 78.54±2.65 | 87.48 |
| PROGRAM | 84.39±1.37 | 79.25±1.67 | 93.83±4.21 | 72.54±0.85 | 82.50 |
| DEYO | 86.31±1.02 | 83.89±0.80 | 96.11±0.49 | 80.21±0.13 | 86.63 |
| CUPLOT | 88.67±0.81 | 87.74±0.58 | 96.61±0.59 | 78.47±3.66 | 87.87 |

Table 21: Full results on the PACS dataset with ResNet-50.

| Methods | A | C | P | S | Avg. |
|---|---|---|---|---|---|
| ERM | 85.24±1.79 | 79.65±2.05 | 96.29±0.68 | 80.71±2.21 | 85.47 |
| BN | 86.51±1.21 | 83.92±1.96 | 96.55±0.39 | 77.37±0.86 | 86.09 |
| TENT | 86.82±1.23 | 84.27±1.89 | 96.61±0.44 | 78.60±0.88 | 86.58 |
| PL | 87.44±1.53 | 82.51±4.43 | 94.79±2.13 | 79.77±2.39 | 86.13 |
| SHOT-IM | 86.34±0.54 | 82.75±1.69 | 94.75±0.30 | 77.55±1.71 | 85.35 |
| T3A | 85.48±2.19 | 81.08±1.13 | 96.79±0.28 | 80.68±2.22 | 86.01 |
| TAST | 87.51±0.94 | 84.09±1.80 | 96.89±0.74 | 77.75±0.96 | 86.56 |
| TAST-BN | 89.18±1.28 | 86.04±1.38 | 97.11±0.81 | 84.57±0.39 | 89.23 |
| TSD | 90.97±0.67 | 90.03±0.99 | 97.42±0.37 | 85.71±0.13 | 91.03 |
| PROGRAM | 87.18±1.38 | 84.26±1.80 | 96.65±0.31 | 77.65±0.70 | 86.44 |
| DEYO | 88.72±0.58 | 85.27±1.61 | 96.79±0.33 | 82.56±0.99 | 88.34 |
| CUPLOT | 91.33±1.15 | 90.00±1.62 | 97.60±0.57 | 85.51±0.65 | 91.11 |

Table 22: Full results on the VLCS dataset with ResNet-18.

| Methods | C | L | S | V | Avg. |
|---|---|---|---|---|---|
| ERM | 96.42±1.37 | 63.79±1.16 | 70.49±1.41 | 70.21±2.53 | 75.23 |
| BN | 82.64±2.50 | 59.22±0.90 | 62.91±2.10 | 70.19±1.34 | 68.74 |
| TENT | 83.23±2.41 | 59.61±0.99 | 63.47±2.18 | 70.70±1.06 | 69.25 |
| PL | 91.92±1.31 | 62.41±1.16 | 69.61±1.20 | 74.44±1.85 | 74.60 |
| SHOT-IM | 89.47±3.48 | 58.85±1.24 | 64.16±2.85 | 71.49±0.69 | 70.99 |
| T3A | 99.32±0.18 | 63.89±1.66 | 69.99±1.90 | 70.51±2.77 | 75.93 |
| TAST | 94.65±1.45 | 55.58±2.32 | 62.78±3.27 | 70.50±1.58 | 70.88 |
| TAST-BN | 97.45±0.76 | 62.58±5.78 | 65.65±0.79 | 74.38±2.34 | 75.02 |
| TSD | 94.35±3.15 | 64.82±1.03 | 66.94±1.48 | 73.11±2.66 | 74.81 |
| PROGRAM | 95.87±1.45 | 59.88±0.61 | 64.08±4.68 | 69.58±0.60 | 72.35 |
| DEYO | 93.47±2.93 | 60.42±6.09 | 68.18±2.60 | 74.13±1.62 | 74.05 |
| CUPLOT | 99.41±0.15 | 65.61±0.50 | 71.25±2.23 | 71.61±1.40 | 76.97 |

Table 23: Full results on the VLCS dataset with ResNet-50.

| Methods | C | L | S | V | Avg. |
|---|---|---|---|---|---|
| ERM | 97.22±0.54 | 64.77±3.09 | 70.95±1.24 | 73.63±0.88 | 76.64 |
| BN | 80.28±2.12 | 58.00±0.22 | 62.29±1.20 | 72.83±0.68 | 68.35 |
| TENT | 81.74±2.03 | 58.37±0.25 | 63.00±1.19 | 73.22±0.56 | 69.08 |
| PL | 91.42±3.13 | 59.29±4.86 | 70.55±0.73 | 73.97±3.08 | 73.81 |
| SHOT-IM | 83.11±5.17 | 57.11±0.62 | 63.12±1.56 | 73.95±0.89 | 69.32 |
| T3A | 98.70±0.82 | 65.79±3.97 | 73.31±2.22 | 71.84±1.03 | 77.41 |
| TAST | 84.99±7.20 | 53.05±1.62 | 63.17±1.33 | 72.91±0.84 | 68.53 |
| TAST-BN | 87.30±1.95 | 58.32±2.49 | 65.51±0.60 | 75.39±0.24 | 71.63 |
| TSD | 92.86±2.13 | 58.39±0.63 | 67.09±2.65 | 76.94±0.60 | 73.82 |
| PROGRAM | 86.83±4.58 | 58.98±0.22 | 56.10±7.43 | 71.77±1.57 | 68.42 |
| DEYO | 83.89±1.47 | 59.96±2.16 | 64.65±3.56 | 73.45±0.64 | 70.49 |
| CUPLOT | 99.18±0.59 | 65.96±1.85 | 74.12±2.62 | 76.50±0.61 | 78.94 |

Table 24: Full results on the OfficeHome dataset with ResNet-18.

| Methods | A | C | P | R | Avg. |
|---|---|---|---|---|---|
| ERM | 55.51±0.41 | 48.93±0.36 | 71.58±0.52 | 73.61±0.46 | 62.41 |
| BN | 54.79±0.32 | 49.41±0.88 | 70.99±0.83 | 73.24±0.46 | 62.11 |
| TENT | 54.95±0.37 | 49.66±0.79 | 71.27±0.89 | 73.33±0.48 | 62.30 |
| PL | 55.16±0.32 | 50.38±0.65 | 71.02±1.32 | 73.58±0.10 | 62.54 |
| SHOT-IM | 56.25±0.64 | 51.59±0.54 | 72.83±0.06 | 73.81±0.47 | 63.62 |
| T3A | 56.04±0.75 | 50.92±0.46 | 73.67±0.41 | 74.70±0.76 | 63.83 |
| TAST | 55.33±0.80 | 50.94±1.25 | 73.96±0.90 | 73.90±1.04 | 63.53 |
| TAST-BN | 54.74±0.38 | 50.36±0.78 | 72.35±0.78 | 71.85±0.23 | 62.33 |
| TSD | 56.90±0.48 | 50.03±1.30 | 72.17±0.97 | 73.38±0.25 | 63.12 |
| PROGRAM | 55.62±0.61 | 50.09±1.83 | 72.07±0.08 | 73.74±0.57 | 62.88 |
| DEYO | 56.38±0.25 | 50.36±0.70 | 71.86±1.08 | 73.60±0.30 | 63.05 |
| CUPLOT | 57.31±0.57 | 52.11±0.80 | 73.96±0.45 | 74.82±0.45 | 64.55 |

Table 25: Full results on the OfficeHome dataset with ResNet-50.

| Methods | A | C | P | R | Avg. |
|---|---|---|---|---|---|
| ERM | 62.93±0.36 | 53.35±0.82 | 76.27±0.09 | 78.21±0.42 | 67.69 |
| BN | 62.67±0.36 | 53.46±0.45 | 75.08±0.66 | 77.52±0.80 | 67.18 |
| TENT | 62.96±0.30 | 54.26±0.39 | 75.18±0.55 | 77.53±0.72 | 67.48 |
| PL | 63.73±0.41 | 55.21±0.60 | 73.64±0.98 | 77.85±0.69 | 67.61 |
| SHOT-IM | 63.59±1.12 | 54.28±0.16 | 75.96±0.41 | 78.10±0.59 | 67.98 |
| T3A | 63.25±0.22 | 54.95±0.85 | 77.79±0.21 | 79.04±0.12 | 68.76 |
| TAST | 63.62±0.27 | 55.46±0.81 | 77.51±0.63 | 78.21±0.59 | 68.70 |
| TAST-BN | 63.78±0.34 | 55.76±0.73 | 76.84±0.49 | 78.01±0.28 | 68.60 |
| TSD | 64.73±0.41 | 57.15±0.55 | 76.78±0.54 | 77.78±0.70 | 69.11 |
| PROGRAM | 63.55±0.79 | 54.27±0.29 | 76.27±1.01 | 77.85±0.77 | 67.99 |
| DEYO | 63.96±0.27 | 55.22±0.91 | 75.96±0.42 | 77.87±0.85 | 68.25 |
| CUPLOT | 66.64±0.69 | 57.87±0.55 | 77.62±0.32 | 79.05±0.15 | 70.30 |

Table 26: Full results on the DomainNet dataset with ResNet-18.

| Methods | clipart | infograph | painting | quickdraw | real | sketch | Avg. |
|---|---|---|---|---|---|---|---|
| ERM | 50.42±0.13 | 15.32±0.15 | 41.83±0.09 | 11.46±0.43 | 51.74±0.34 | 43.64±0.21 | 35.74 |
| BN | 50.75±0.11 | 11.26±0.21 | 40.71±0.18 | 11.12±0.12 | 51.86±0.30 | 43.70±0.23 | 34.90 |
| TENT | 51.16±0.12 | 12.47±0.23 | 41.84±0.25 | 10.65±0.33 | 51.28±0.20 | 44.76±0.17 | 35.36 |
| PL | 50.88±0.06 | 13.16±0.37 | 41.19±0.15 | 10.69±0.57 | 51.72±0.42 | 44.02±0.26 | 35.28 |
| SHOT-IM | 50.90±0.13 | 12.76±0.39 | 41.36±0.18 | 13.58±0.11 | 52.37±0.30 | 44.38±0.23 | 35.89 |
| T3A | 50.36±0.29 | 15.14±0.15 | 40.26±0.05 | 16.22±0.19 | 53.02±0.13 | 42.74±0.26 | 36.29 |
| TAST | 50.43±0.27 | 10.67±0.05 | 40.69±0.09 | 14.22±0.19 | 53.69±0.33 | 42.49±0.26 | 35.37 |
| TAST-BN | 50.12±0.31 | 11.32±0.13 | 40.82±0.19 | 14.11±0.29 | 52.11±0.21 | 42.19±0.31 | 35.11 |
| TSD | 50.75±0.13 | 11.71±0.13 | 42.35±1.17 | 11.96±0.67 | 52.03±0.33 | 44.20±0.21 | 35.50 |
| PROGRAM | 50.95±0.08 | 13.09±0.33 | 41.67±0.15 | 13.28±0.24 | 52.35±0.34 | 44.27±0.25 | 35.94 |
| DEYO | 50.85±0.05 | 13.23±0.25 | 41.20±0.18 | 10.99±0.17 | 51.89±0.33 | 43.98±0.27 | 35.36 |
| CUPLOT | 51.77±0.11 | 14.95±0.06 | 42.29±0.11 | 15.89±0.27 | 54.48±0.28 | 44.72±0.24 | 37.35 |

Table 27: Full results on the DomainNet dataset with ResNet-50.

| Methods | clipart | infograph | painting | quickdraw | real | sketch | Avg. |
|---|---|---|---|---|---|---|---|
| ERM | 61.14±0.23 | 20.89±0.23 | 49.74±0.29 | 13.68±0.29 | 62.08±0.20 | 52.20±0.41 | 43.29 |
| BN | 60.58±0.23 | 15.19±0.12 | 48.66±0.12 | 11.95±0.24 | 61.18±0.26 | 51.66±0.15 | 41.54 |
| TENT | 61.71±0.24 | 17.36±0.09 | 50.33±0.13 | 10.26±0.77 | 61.58±0.18 | 53.27±0.08 | 42.42 |
| PL | 61.04±0.22 | 17.62±0.43 | 49.93±0.06 | 11.75±0.47 | 61.37±0.17 | 52.59±0.19 | 42.38 |
| SHOT-IM | 61.40±0.39 | 17.51±0.09 | 49.82±0.13 | 16.54±0.53 | 62.65±0.18 | 52.81±0.21 | 43.46 |
| T3A | 61.13±0.34 | 21.01±0.18 | 48.82±0.11 | 18.67±0.49 | 63.32±0.15 | 51.69±0.33 | 44.11 |
| TAST | 60.77±0.42 | 14.95±0.20 | 48.96±0.14 | 15.16±0.27 | 62.85±0.36 | 51.56±0.18 | 42.38 |
| TAST-BN | 60.89±0.29 | 15.31±0.25 | 48.99±0.09 | 14.92±0.23 | 62.98±0.28 | 51.83±0.19 | 42.49 |
| TSD | 60.80±0.29 | 15.52±0.11 | 49.42±0.08 | 13.88±0.24 | 61.70±0.19 | 52.28±0.18 | 42.27 |
| PROGRAM | 61.15±0.26 | 18.05±0.08 | 49.99±0.28 | 15.48±0.40 | 62.23±0.15 | 53.22±0.21 | 43.35 |
| DEYO | 61.03±0.21 | 18.05±0.32 | 49.89±0.09 | 12.00±0.25 | 61.33±0.13 | 52.51±0.21 | 42.47 |
| CUPLOT | 62.34±0.19 | 20.76±0.18 | 50.48±0.01 | 18.60±0.52 | 64.57±0.19 | 53.15±0.33 | 44.98 |

