# OpenReview forum: "Toward Fine-Grained Domain Knowledge: Curriculum Pseudo-Labeling for Online Test-Time Adaptation"
_ICLR.cc/2026/Conference — Submitted to ICLR 2026_

### Official Review · Reviewer_9R4p · 2025-10-25

**Soundness:** 2
**Presentation:** 2
**Contribution:** 2
**Rating:** 4
**Confidence:** 4

**Summary:**

This paper presents CUPLOT, a curriculum-learning-based framework for achieving fine-grained online test-time adaptation.
The proposed method facilitates the absorption of fine-grained domain knowledge mainly through batch-level curriculum partitioning.

**Strengths:**

1. The paper introduces the concept of curriculum learning into the Online Test-Time Adaptation (OTTA) setting, re-examining model adaptation from the perspective of fine-grained domain knowledge absorption and providing a novel and inspiring research viewpoint.

2. The proposed framework is independent of specific network architectures or loss functions, allowing it to be easily integrated with various existing OTTA methods and demonstrating good scalability and generality.

3. The authors provide extensive benchmark experiments to support the effectiveness of the proposed approach.

**Weaknesses:**

1.	The motivation for fine-grained domain knowledge learning is intuitively reasonable. However, the authors still need to provide a solid experimental or theoretical foundation rather than relying solely on descriptive explanations of this motivation.


2.	The core procedure of CUPLOT appears to rely on gradient-based curriculum selection. Is there any further justification for the rationality of this selection mechanism? Could traditional confidence-based or feature-level selection strategies also be considered? What specific advantages does CUPLOT offer compared to them?


3.	In the theoretical analysis, some assumptions are relatively strong (e.g., assuming that pseudo-label errors are strictly correlated with μ ranking). Moreover, the analysis does not quantitatively measure the improvement of fine-grained curriculum over convergence speed, making it more of a “plausibility argument” rather than a rigorous mathematical derivation.

4.	Most of the compared methods are batch-level OTTA baselines. Theoretically, the proposed method could also be extended to the Continual TTA setting. Including such experiments would make the work more comprehensive and convincing.

5.	The algorithmic description of CUPLOT is somewhat complex and confusing, while the explanation itself is rather brief. It would be helpful if the authors could provide a methodological framework figure to give readers a clearer and more intuitive understanding of the approach.

**Questions:**

Please refer to Weaknesses.

---

> ### Author Response · Authors · 2025-11-21
> **Responses to Reviewer 9R4p (Part 1 W1, W2)**
>
> Thank you for your thoughtful review. We are glad to address your concerns below.
>
> > **W1:** "The motivation for fine-grained domain knowledge learning is intuitively reasonable. However, the authors still need to provide a solid experimental or theoretical foundation rather than relying solely on descriptive explanations of this motivation."
>
> We thank the reviewer for noting that *“the motivation for fine-grained domain knowledge learning is intuitively reasonable”* and for the helpful suggestion to further substantiate it. We provide both **experimental evidence** and **theoretical justification** supporting the distinction between coarse-grained and fine-grained domain knowledge in CUPLOT.
>
> 1. **Experimental Evidence**: We conduct an additional experiment comparing four learning orders: (1) our gradient-consistency curriculum order, (2) a random order, (3) an intentionally poor order obtained by reversing the ranked scores, and (4) ordinary batch. The adaptation accuracy curves across the test stream show that our ordering consistently achieves higher accuracy than other cases under identical settings, demonstrating that the order in which samples are learned affects how effectively fine-grained domain knowledge is absorbed during online adaptation. The corresponding figure will appear on **Page 2 of the revised version**, and the related discussion will be **incorporated into the Introduction** to make the motivation clearer.
>
> 2. **Theoretical Justification**: Section 3.3 (Theorem 1) provides a formal explanation of why learning order matters. The analysis decomposes the pseudo-label error into a coarse-aligned term and a fine-grained residual term whose influence accumulates across adaptation steps. The theorem shows that ordering samples to minimize gradient-consistency discrepancy yields a strictly tighter upper bound on this residual error, thereby reducing its propagation along the adaptation trajectory. This theoretically supports the intuition that learning coarse-grained structure first and progressively incorporating fine-grained knowledge leads to more superior adaptation.
>
>
> > **W2:** "The core procedure of CUPLOT appears to rely on gradient-based curriculum selection. Is there any further justification for the rationality of this selection mechanism? Could traditional confidence-based or feature-level selection strategies also be considered? What specific advantages does CUPLOT offer compared to them?"
>
> The contribution of our method lies in enabling the model to **progressively absorb fine-grained domain knowledge**, which is a noteworthy point under domain shift in test-time. We use a gradient-consistency–driven curriculum learning strategy as the facilitating mechanism.
>
> We agree that it is valuable to clarify why CUPLOT adopts a gradient-based curriculum rather than other strategies such as confidence-based or feature-level selection strategies. We have shown that gradient consistency is a more effective indicator of sample difficulty than entropy-based strategies in Table 12 and 13. Here, as suggested, we empirically conduct an ablation comparing gradient consistency with another two alternatives:
> (1) **Softmax Confidence**: For sample $x_i$, we compute the predicted class probabilities $q_i = \mathrm{softmax}(f_\theta(x_i))$. The confidence score is defined as $s_i = \max_{j} q_i^{j}$. Samples with lower confidence are considered harder and are scheduled later in the curriculum.
> (2) **Feature-space Similarity (Prototype-based Selection)**: For each class $j$, we maintain a prototype vector $w_j$, typically computed as the mean feature representation of samples in the batch. For sample $x_i$, we extract its feature $z_i$ and compute its distance to the nearest class prototype: $s_i = \min_{j}\| \phi_\theta(x_i) - w_{j} \|_2$. Samples with larger distances are considered harder and scheduled later in the curriculum, while samples closer to the prototype are scheduled earlier.
>
> The results are reported in Table C. From the tables, we validate the effectiveness of gradient consistency compared to confidence-based and feature-level selection strategies.
>
> **Table C.** Classification accuracy of different consistency metrics on CIFAR-100-C and DomainNet.
> |Methods|CIFAR-100-C|DomainNet|
> |---|---|---|
> |Ours|**60.11**|**37.35**|
> |Confidence|59.87|37.23|
> |Feature|59.72|37.07|
>
> **Rationality of gradient-based selection.**
>
> As mentioned in lines 53 of the Introduction, if the gradient of an instance is more inconsistent with the overall gradient on the batch, its domain knowledge will be diluted or even harm the absorption of domain knowledge
> from other instances, leading to less knowledge being absorbed at the batch level. Instances with high consistency in gradient consistency reinforce the dominant gradient signal and typically correspond to easier or more stable samples, while low-consistency instances are harder or more uncertain.

---

> ### Author Response · Authors · 2025-11-21
> **Responses to Reviewer 9R4p (Part 1 W3)**
>
> > **W3:** "In the theoretical analysis, some assumptions are relatively strong (e.g., assuming that pseudo-label errors are strictly correlated with μ ranking). Moreover, the analysis does not quantitatively measure the improvement of fine-grained curriculum over convergence speed, making it more of a “plausibility argument” rather than a rigorous mathematical derivation."
>
> Thank you for raising this point. We would first like to clarify that our analysis does not assume a strict or perfect monotonic relationship between pseudo-label errors and the $\mu$ ranking. The $\mu$ score serves only as a relative difficulty estimate: instances with more reliable pseudo-labels are expected to appear earlier in the curriculum, on average. This is a mild and intuitive assumption that aligns with common practice in prior test-time adaptation methods. In fact, prior TTA approaches [1,2,3], particularly the theoretical analysis in [3], rely on sample selection or filtering and implicitly make the same assumption: pseudo-labels associated with certain measures (e.g., lower entropy) tend to be more reliable. Our formulation simply makes this implicit assumption explicit and uses it to articulate why instance-level ordering mitigates pseudo-label error amplification.
>
> [1] Wang et al. Feature Alignment and Uniformity for Test-Time Adaptation. CVPR, 2023. [2] Jang et al. Test-Time Adaptation via Self-Training with Nearest Neighbor Information. ICLR, 2023. [3] Gui, and Li et al. Active Test-Time Adaptation: Theoretical Analyses and an Algorithm. ICLR, 2024.
>
> Additionally, other assumptions are not intended to be strong and they are Lipschitz-type assumptions used purely to formalize an intuitive mechanism:
>
> *1. "(Assumption 1.) Let $\begin{array} { r } { \nabla \Theta ( \mathcal { D } ) = \sum _ { i \in \mathcal { D } } \alpha \frac { \partial \ell ( f ( x ; \Theta ) , d _ { i } ) } { \partial \Theta } } \end{array}$ denote the gradient of the model $\begin{array} { r } f (·; \Theta) \end{array}$ using pseudo-labels on the instances with any index set $\mathcal { D }$ . Then there exists the constant $\zeta > 0$ , we have $\begin{array} { r } { | | \Theta - \nabla \Theta ( \mathcal { D } ) - \Theta ^ { \star } | | \leq \zeta \frac { | \tilde { \mathcal { I } } \cap \mathcal { D } | } { | \mathcal { I } \cap \mathcal { D } | } } \end{array}$."*
>
> Assumption 1 simply says that the deviation between the updated parameter ( after using pseudo-labels on a set $\mathcal { D }$ ) and the ground-truth optimum $\Theta ^ { \star }$ is proportional to the pseudo-label error ratio in that set.
>
> Why this is a standard and intuitive assumption: (1) It is a **Lipschitz-type bound**, commonly assumed in pseudo-label learning analysis. (2) It does *not* require a strict or deterministic relation, only that **more pseudo-label errors cause proportionally larger gradient deviation**. (3) This matches common intuition: If pseudo-labels contain fewer errors, gradients move parameters closer to $\Theta^{\star}$.
>
>
> *2. (Assumptions in Theorem 1) There exist the constants $\beta , \gamma > 0$ $| f _ { j } ( x ; \Theta ) - p ( j | x ) | \le \beta | | \Theta - \Theta ^ { \star } | |$ and $\frac { | \tilde { \mathcal { I } } \cap \mathcal { D } | } { | \mathcal { I } \cap \mathcal { D } | } \leq \gamma | | \Theta - \Theta ^ { \star } | |$.*
>
>
> Meaning: The model’s prediction error is **Lipschitz in parameter space**, which is standard in convergence analyses of deep models and pseudo-label learning. And the pseudo-label error ratio decreases as the model approaches the optimum $\Theta ^ { \star }$.
>
> Why this is a standard and intuitive assumption: (1) They encode only that training improves pseudo-label quality, which empirically holds for all pseudo-label methods. (2) They merely allow us to express improvement curves using proportional relationships.
>
> ##### About Convergence
>
> In our paper, the quantity $ e^{\star} $ is explicitly defined to reflect the model's convergence to the Bayes-optimal classifier during the TTA process. A smaller $ e^* $ corresponds to closer alignment with the Bayes-optimal classifier. Importantly, in the proof of Theorem 1, we show that the pseudo-label error $ e^{\star}\prime $ obtained under our curriculum-based framework satisfies $ e^{\star}\prime \le e^{\star} $, where $ e^{\star} $ corresponds to the batch-level baseline. This inequality rigorously demonstrates that our instance-level curriculum reduces the effective pseudo-label error, and thus the model under our framework converges to the Bayes-optimal classifier at a faster rate than the batch-level method. We will also emphasize this point explicitly following Theorem 1 in the revised version to make the improved convergence clear. Thank you for your helpful reminder.

---

> > ### Comment · Reviewer_9R4p · 2025-11-23
> >
> > After carefully reviewing the authors’ rebuttal, I find that they have addressed most of my concerns, especially those related to the theoretical explanations. I will accordingly raise my score.

---

> > > ### Author Response · Authors · 2025-11-23
> > >
> > > Dear Reviewer 9R4p,
> > >
> > > Thank you sincerely for your confirmation that our responses have addressed most of your concerns. We greatly appreciate your time and constructive feedback throughout the review process, which have been instrumental in refining our work.
> > >
> > > Sincerely,
> > >
> > > Paper 15203 Authors

---

> ### Author Response · Authors · 2025-11-21
> **Responses to Reviewer 9R4p (Part 1 W4, W5)**
>
> > **W4:** "Most of the compared methods are batch-level OTTA baselines. Theoretically, the proposed method could also be extended to the Continual TTA setting. Including such experiments would make the work more comprehensive and convincing."
>
> We appreciate the reviewer’s suggestion that our approach could be extended to the Continual TTA setting. Following the protocol in [1], we conduct additional experiments comparing CUPLOT with two representative baselines [1,2]. For a fair comparison, we integrate an EMA teacher into our framework. The results in Table B show that the curriculum-driven ordering in CUPLOT continues to provide consistent improvements in the Continual TTA setting. These comparisons and discussions will be included in the revised version.
>
> Table B. Classification **error rate** (%) on the CIFAR100-to-CIFAR100C online continual test-time adaptation task under the highest corruption severity level (severity 5), evaluated on ResNeXt-29 following the [1] protocol.
>
> |Methods|Gaussian|shot|impulse|defocus|glass|motion|zoom|snow|frost|fog|brightness|contrast|elastic|pixelate|jpeg|Mean|
> |---|---|---|---|---|---|---|---|---|---|---|---|---|---|---|---|---|
> |Source|73.0|68.0|39.4|29.3|54.1|30.8|28.9|39.5|45.8|50.3|29.5|55.1|37.2|74.7|41.2|46.4|
> |BN|42.1|40.7|42.7|27.6|41.9|29.7|27.9|34.9|35.0|41.5|26.5|30.3|35.7|32.9|41.2|35.4|
> |TENT|37.2|35.8|41.7|37.9|51.2|48.3|45.8|58.4|63.7|71.1|70.4|82.3|88.0|88.5|90.4|60.9|
> |CoTTA|40.1|37.7|39.7|26.9|38.0|27.9|26.4|32.8|31.8|40.3|24.7|26.9|32.5|28.3|33.5|32.5|
> |RMT|38.5|34.4|35.4|26.4|32.7|27.0|25.0|27.6|27.5|30.0|24.0|25.8|27.0|25.2|28.4|29.0|
> |Ours|38.0|34.0|34.8|26.0|32.2|26.4|24.5|27.2|27.1|29.5|23.5|25.3|26.6|24.7|27.9|28.5|
>
> [1] Wang et al. Continual test-time domain adaptation. CVPR. 2022.
> [2] Döbler et al. Robust mean teacher for continual and gradual test-time adaptation. CVPR. 2023.
>
> > **W5:** "The algorithmic description of CUPLOT is somewhat complex and confusing, while the explanation itself is rather brief. It would be helpful if the authors could provide a methodological framework figure to give readers a clearer and more intuitive understanding of the approach."
>
> Thank you for the suggestion. We will add a clear framework figure of CUPLOT in the revised version (Page 2, Figure 1(a)) to illustrate the overall pipeline and improve readability.

---

### Official Review · Reviewer_tWX1 · 2025-10-26

**Soundness:** 3
**Presentation:** 2
**Contribution:** 2
**Rating:** 4
**Confidence:** 4

**Summary:**

This paper proposes CUPLOT, a curriculum-learning-based framework
designed to achieve fine-grained Online Test-Time Adaptation (OTTA).
By introducing a gradient-consistency-driven curriculum scheduling strategy,
CUPLOT effectively enhances the model’s ability to absorb target-domain knowledge.
Furthermore, a theoretical analysis of its convergence property is provided,
demonstrating the robustness and potential advantages of the proposed method.

**Strengths:**

1. The paper takes fine-grained domain knowledge under the OTTA setting as its starting point, which
is an intuitively reasonable and meaningful motivation for enhancing adaptation granularity.

2. Theoretical analysis and extensive benchmark experiments consistently verify the effectiveness
and robustness of the proposed CUPLOT framework.

**Weaknesses:**

1. The core motivation of CUPLOT lies in distinguishing between coarse-grained and fine-grained domain knowledge learning. However, the current discussion remains largely descriptive and is not sufficiently convincing. It is recommended that the authors include corresponding experiments or theoretical analysis when introducing the motivation to strengthen its persuasiveness.

2. The key component of CUPLOT lies in its
curriculum division mechanism, which essentially determines the learning sequence of samples within each batch. I would like the authors to provide a comparison between different sequence selection strategies (e.g., random selection, sequential selection, or other sequence-based sampling approaches) to better demonstrate the necessity and effectiveness of the proposed scheduling method.

3. The mathematical formulas and notations are rather confusing, which makes the paper appear overly engineering-oriented and difficult to follow. A clearer presentation of equations and symbols is strongly encouraged.

4. I am confused about the description in line 204, where $B = \operatorname{round}\big(\log(1 - s_{t,k-1})\big)$ and $K^t$ is set around $ \log n_t $ . If $B$ is dynamically varying while  is approximately fixed, would this inconsistency cause conflicts or ambiguity in the curriculum scheduling algorithm?

5. Regarding Equation (6), if only a few unlearned samples remain in the batch, then $g_i^{t,k}$ and $G^{t,k}$ obtained from Equations (4) and (5) would be almost identical. In that case, the gradient consistency term $\mu_i^{t,k}$could approach infinity. How do the authors handle this situation in practice?

6. The paper lacks visual illustrations such as a framework or methodological diagram, which significantly affects readability. It is suggested that the authors include a clear framework figure to help readers better understand the overall pipeline of CUPLOT.

**Questions:**

Please refer to the Weaknesses.

---

> ### Author Response · Authors · 2025-11-21
> **Responses to Reviewer tWX1 (Part 1 W1, W2, W3)**
>
> Thank you for your detailed technical comments. In response to the weaknesses, we would like to provide the following explanations.
>
> > **W1:** "The core motivation of CUPLOT lies in distinguishing between coarse-grained and fine-grained domain knowledge learning. However, the current discussion remains largely descriptive and is not sufficiently convincing. It is recommended that the authors include corresponding experiments or theoretical analysis when introducing the motivation to strengthen its persuasiveness."
>
> Thank you for helping us make the motivation more convincing. We provide both **experimental evidence** and **theoretical justification** supporting the distinction between coarse-grained and fine-grained domain knowledge in CUPLOT. The new experiment directly shows that the proposed gradient-consistency curriculum produces consistently stronger adaptation behavior than random or intentionally suboptimal learning orders. Complementarily, our theoretical analysis formalizes why ordering affects the accumulation of fine-grained residual errors during online adaptation.
>
> 1. **Experimental Evidence**: We conduct an additional experiment comparing four learning orders on CIFAR-100-C: (1) our gradient-consistency curriculum order, (2) a random order, (3) an intentionally poor order obtained by reversing our curriculum ranking, and (4) ordinary batch. The adaptation accuracy curves across the test stream show that our ordering consistently achieves higher accuracy than other cases under identical settings, demonstrating that the order in which samples are learned affects how effectively fine-grained domain knowledge is absorbed during online adaptation. The corresponding figure will appear on **Page 2 of the revised version**, and the related discussion will be **incorporated into the Introduction** to make the motivation clearer.
>
> 2. **Theoretical Justification**: Section 3.3 (Theorem 1) provides a formal explanation of why learning order matters. The analysis decomposes the pseudo-label error into a coarse-aligned term and a fine-grained residual term whose influence accumulates across adaptation steps. The theorem shows that ordering samples to minimize gradient-consistency discrepancy yields a strictly tighter upper bound on this residual error, thereby reducing its propagation along the adaptation trajectory. This theoretically supports the intuition that learning coarse-grained structure first and progressively incorporating fine-grained knowledge leads to superior adaptation.
>
> > **W2:** "The key component of CUPLOT lies in its curriculum division mechanism, which essentially determines the learning sequence of samples within each batch. I would like the authors to provide a comparison between different sequence selection strategies (e.g., random selection, sequential selection, or other sequence-based sampling approaches) to better demonstrate the necessity and effectiveness of the proposed scheduling method."
>
> As you suggested, to validate the effectiveness of our curriculum-based scheduling, we compare it with **random selection** and **sequential (input-order) selection** under the same test-time adaptation setup. Results are shown in Table A.
>
> **Table A. Accuracy of different sequence strategies.**
>
> | Methods              | CIFAR-10-C | CIFAR-100-C | ImageNet-C |
> | -------------------- | ---------- | ----------- | ---------- |
> | sequential selection | 86.52      | 58.54       | 44.01      |
> | random selection     | 86.47      | 58.50       | 44.06      |
> | CUPLOT        | **87.35**  | **60.11**   | **55.21**  |
>
> Both random and sequential orders achieve similar performance, whereas our gradient-consistency–driven curriculum consistently improves accuracy. This demonstrates that CUPLOT’s gains are not due to arbitrary ordering but result from the proposed principled curriculum. We will include this table and discussion in the revision Appendix A.9.
>
>
> > **W3:** "The mathematical formulas and notations are rather confusing, which makes the paper appear overly engineering-oriented and difficult to follow. A clearer presentation of equations and symbols is strongly encouraged."
>
> We thank the reviewer for the suggestion regarding the clarity of our mathematical presentation. We have polished key equations and their surrounding notation in the revised version to improve readability and reduce potential confusion. These changes aim to make the theoretical derivations more accessible while keeping the overall structure intact.

---

> ### Author Response · Authors · 2025-11-21
> **Responses to Reviewer tWX1 (Part 2 W4, W5, W6)**
>
> > **W4:** "I am confused about the description in line 204, where B = round( log(1 - s_{t,k-1}) ) and K^t is set around log n_t . If B is dynamically varying while is approximately fixed, would this inconsistency cause conflicts or ambiguity in the curriculum scheduling algorithm?"
>
> We thank the reviewer for the comment. To clarify, $ K^t $ denotes the number of curriculum steps within a batch, while $ B $ indicates the number of samples selected in each curriculum step. These two quantities operate at different levels and are **not contradictory**. In fact, $ B $ is chosen so that after $ K^t $ curriculum steps, all samples in the batch are covered, ensuring a complete and well-structured curriculum. This design guarantees that the curriculum scheduling is consistent and unambiguous.
>
> > **W5:** "Regarding Equation (6), if only a few unlearned samples remain in the batch, then g_{i}^{t,k} and G^{t,k} obtained from Equations (4) and (5) would be almost identical. In that case, the gradient consistency term μ_{i}^{t,k} could approach infinity. How do the authors handle this situation in practice?"
>
> We thank the reviewer for pointing this out. In practice, when $k-1$ curriculum steps have selected samples, the remaining samples in the $ k$-th step are simply all the unselected samples, and no further selection is needed. In this case, the gradient consistency term $ \mu_{i}^{t,k} $ may indeed approach infinity, but this has **no impact on the algorithm**, as the final curriculum step deterministically includes all remaining samples. We will clarify this handling in the revised version.
>
> > **W6:** "The paper lacks visual illustrations such as a framework or methodological diagram, which significantly affects readability. It is suggested that the authors include a clear framework figure to help readers better understand the overall pipeline of CUPLOT."
>
> Thank you for the suggestion. We will add a clear framework figure of CUPLOT in the revised version (Page 2, Figure 1(b)) to illustrate the overall pipeline and improve readability.

---

> ### Author Response · Authors · 2025-11-27
>
> Dear Reviewer tWX1,
>
> We greatly appreciate the time and effort you've taken to review our submission. We hope that our response has addressed the concerns raised in your initial reviews, and we look forward to your feedback.
>
> Please let us know if you need any further information or clarification.
>
> Best regards and thanks,
>
> Paper 15203 Authors

---

### Official Review · Reviewer_pezg · 2025-10-31

**Soundness:** 2
**Presentation:** 3
**Contribution:** 3
**Rating:** 6
**Confidence:** 4

**Summary:**

CUPLOT (Curriculum Pseudo-Labeling via Ordered Training) introduces a curriculum-learning framework for online test-time adaptation. Rather than updating the model on each mini-batch as a single block, CUPLOT divides each batch into several smaller curricula and orders the instances so that the model learns easier or more reliable samples first and harder or noisier ones later. The paper further demonstrates that curriculum pseudo-labeling enhances the model’s adaptation ability and provides a tighter theoretical bound toward the Bayes-optimal classifier on the target domain. Extensive experiments validate the effectiveness of the proposed framework across multiple benchmarks

**Strengths:**

- The paper is clearly written and well-structured, allowing readers to easily follow the logical flow and understand the motivation and proposed method at both conceptual and technical levels.

- The empirical results convincingly demonstrate the effectiveness of the proposed approach across multiple standard benchmark datasets.

**Weaknesses:**

- **Limited experimental scope:** The experiments are not sufficiently comprehensive. Since the paper’s main contribution lies in introducing a conceptual framework for within-batch curriculum learning, it is largely orthogonal to prior methods. The paper could conduct more extensive experiments by integrating the proposed curriculum mechanism with existing approaches to demonstrate its general applicability and potential performance gains.

- **Insufficient emphasis on curriculum learning in the Introduction and Method:** The discussion of curriculum learning could be elaborated further. For instance, in __Line 57__, when introducing the concept, the authors could briefly define the notions of __easy__ and __hard__ instances and clarify how gradient consistency is used to distinguish them within the proposed framework. Additionally, Section 3.1 would benefit from a short subsection that formally presents the formulation of curriculum learning and its relevance to the test-time adaptation setting.

- **Unstructured related work section:** The related work currently lacks clear organization, making it difficult for readers to follow the connections between topics. It is recommended to divide the section into three subsections—__(i) problem setting, (ii) existing methods, and (iii) curriculum learning__—to improve clarity. Moreover, the discussion could be strengthened by providing a deeper conceptual and technical comparison with prior works that incorporate curriculum learning.

**Questions:**

- Could you provide additional experiments or discussion to validate whether CUPLOT consistently improves the performance of other baselines when integrated?

- How do you formally define easy and hard instances in your curriculum scheduling, and what is the rationale behind using gradient consistency as the ordering criterion?

- Can you provide a clearer comparison highlighting what aspects of curriculum learning are novel or distinct in your formulation?

- Could you restructure the related work section to clearly separate discussions of (i) problem setting, (ii) adaptation methods, and (iii) curriculum learning?

---

> ### Author Response · Authors · 2025-11-21
> **Responses to Reviewer pezg (Part 1 W1, W2, Q1, Q2)**
>
> Thank you for your positive feedback and detailed suggestions. In response to weaknesses and questions, we would like to provide the following explanations.
>
> > **W1 & Q1:** "Limited experimental scope: The experiments are not sufficiently comprehensive. Since the paper’s main contribution lies in introducing a conceptual framework for within-batch curriculum learning, it is largely orthogonal to prior methods. The paper could conduct more extensive experiments by integrating the proposed curriculum mechanism with existing approaches to demonstrate its general applicability and potential performance gains." "Could you provide additional experiments or discussion to validate whether CUPLOT consistently improves the performance of other baselines when integrated?"
>
> Thank you for the insightful suggestion. We agree that the proposed curriculum mechanism is conceptually orthogonal to some existing TTA techniques, and demonstrating its general applicability is valuable.
>
> To address this, we have conducted additional experiments integrating our gradient-consistency curriculum with two representative baselines: **SHOT**[1] and **DeYO**[2]. In both cases, we replace their batch-level pseudo-labeling step with CUPLOT’s curriculum-based ordering while keeping all other components unchanged. The results in Table A support the reviewer’s hypothesis that the proposed curriculum is a general and complementary mechanism that enhances other TTA methods—not tied to our prototype-based instantiation.
>
> **Table A**. Accuracy improvements when integrating CUPLOT’s curriculum into SHOT and DeYO across corruption benchmarks.
>
> |Methods|CIFAR-10-C|CIFAR-100-C|ImageNet-C|
> |---|---|---|---|
> |SHOT-IM|86.33|59.14|54.43|
> |SHOT-IM+Ours|**86.95**|**59.87**|**55.25**|
> |DeYO|86.14|59.08|50.43|
> |DeYO+Ours|**87.01**|**59.60**|**51.53**|
>
> We will include these results and an expanded discussion in the revised version to more clearly demonstrate the broad applicability of CUPLOT.
>
> > **W2 & Q2:** "Insufficient emphasis on curriculum learning in the Introduction and Method: The discussion of curriculum learning could be elaborated further. For instance, in Line 57, when introducing the concept, the authors could briefly define the notions of easy and hard instances and clarify how gradient consistency is used to distinguish them within the proposed framework. Additionally, Section 3.1 would benefit from a short subsection that formally presents the formulation of curriculum learning and its relevance to the test-time adaptation setting." "How do you formally define easy and hard instances in your curriculum scheduling, and what is the rationale behind using gradient consistency as the ordering criterion?"
>
> We add more connections to curriculum learning in both the Introduction and Section 3.1. And we revise the manuscript to explicitly introduce easy and hard instances, especially around Eq. (6) in our revision, and to clarify how gradient consistency underpins our curriculum scheduling.
>
> In our formulation, an instance is considered easy if its gradient direction is highly aligned with the model’s current update direction, i.e., it induces a small gradient deviation from its batch neighbors, indicating that the model already captures the underlying pattern well. Conversely, an instance is hard if its gradient direction deviates substantially from others in the batch, suggesting higher uncertainty or potential distributional shift. We quantify this via gradient consistency, computed as the average pairwise gradient similarity between the instance and other samples in the mini-batch.
>
> The rationale for using gradient consistency is twofold:
> (1) **Optimization-based motivation**: samples whose gradients agree with the dominant update direction contribute stable improvements and are less likely to introduce noisy or harmful pseudo-labels, making them suitable to learn from earlier in the curriculum;
> (2) **Theoretical support**: our analysis in Sec. 3.4 shows that learning from gradient-consistent (easy) instances first provides a tighter bound on the error propagation term during online pseudo-labeling.
>
> [1] Liang et al. Do we really need to access the source data? source hypothesis transfer for unsupervised domain adaptation. ICML 2020.
>
> [2] Lee et al. Entropy is not enough for test-time adaptation: From the perspective of disentangled factors. ICLR, 2024.

---

> ### Author Response · Authors · 2025-11-21
> **Responses to Reviewer pezg (Part 2 W3, Q3, Q4)**
>
> > **W3 & Q3 & Q4:** "Unstructured related work section: The related work currently lacks clear organization, making it difficult for readers to follow the connections between topics. It is recommended to divide the section into three subsections—(i) problem setting, (ii) existing methods, and (iii) curriculum learning—to improve clarity. Moreover, the discussion could be strengthened by providing a deeper conceptual and technical comparison with prior works that incorporate curriculum learning." "Can you provide a clearer comparison highlighting what aspects of curriculum learning are novel or distinct in your formulation?" "Could you restructure the related work section to clearly separate discussions of (i) problem setting, (ii) adaptation methods, and (iii) curriculum learning?"
>
> 1. **Restructuring**:
>
> In the revised version, we will reorganize this section into three subsections as you suggest:
> (i) **Problem Setting** (online/streaming TTA),
> (ii) **Adaptation Methods** (entropy-based, prototype-based, MT-based, contrastive, etc.), and
> (iii) **Curriculum Learning** (classical CL, self-paced learning, etc.).
>
> 2. **Comparison with prior works that incorporate curriculum learning**:
>
> Classical curriculum learning defines difficulty using labels, loss values, or manually designed notions of sample easiness. These concepts do not extend naturally to the unlabeled and distribution-shifted TTA setting. In contrast, CUPLOT introduces a **gradient-consistency–based difficulty measure**, which is (a) label-free, (b) stable under shift, and (c) directly grounded in optimization dynamics. Moreover, unlike prior pseudo-label curricula, CUPLOT performs **intra-batch progressive prototype updating**, creating a causal, instance-level curriculum within each batch, which is not present in previous curriculum-learning or TTA approaches.

---

> ### Author Response · Authors · 2025-11-28
>
> Dear Reviewer pezg,
>
> We greatly appreciate the time and effort you've taken to review our submission. We hope that our response has addressed the concerns raised in your initial reviews, and we look forward to your feedback.
>
> Please let us know if you need any further information or clarification.
>
> Best regards and thanks,
>
> Paper 15203 Authors

---

### Official Review · Reviewer_74MR · 2025-11-01

**Soundness:** 3
**Presentation:** 3
**Contribution:** 2
**Rating:** 4
**Confidence:** 3

**Summary:**

The paper proposes CUPLOT, an OTTA framework that leverages instance-aware fine-grained knowledge. It first splits each arriving test data batch into a sequence of curricula ordered by gradient-consistency score, and learns each curriculum using prototype-based pseudo-labels computed from features of previously learned instances. A theoretical analysis argues that curriculum pseudo-labels yield a tighter bound than batch-level pseudo-labels. Experiments on synthetic corruption benchmarks and domain-generalization datasets report small but consistent gains over recent OTTA baselines.

**Strengths:**

1. **Fine-grained OTTA framing.** Different from prior approaches, converting a batch into ordered curricula based on per-instance gradient consistency is a well-designed way to address gradient conflict within a batch. The mechanism is easy to follow.
2. **Active-TTA angle.** The CUPLOT-AT provides a unique angle shows how curriculum scores could drive selective querying under mixed severities. It shows potential for boarder impacts.
3. **Broad empirical sweep. ** The paper evaluates across common OTTA corruption and DG benchmarks, and reports sensitivity to temperature τ, batch size, and curriculum number. CUPLOT shows consistency across different settings and scenarios.

**Weaknesses:**

1.  **Computational overhead not fully accounted.**  CUPLOT requires per-instance gradients to rank samples and then multiple updates per batch across curricula, while TTA is very sensitive with compute overhead since it directly deployed in test time. The appendix varies \(K^t\) and reports time/accuracy trade-offs, but a clear backward-passes-per-sample accounting and wall-clock time comparison vs. strong baselines is missing in the main text.
2. **Incremental combination rather than a new principle.**  Prototypes for pseudo-labels are well-trodden ideas. The novelty mainly lies in the specific gradient-consistency-driven ordering with prototype soft labels.
3.  **Effect sizes are small; significance not emphasized.**  The gains over comparing methods on the three corruption suites are sub-1% on average. It’s hard to tell its advantages.
4. **Limited comparison to strong Mean-teacher-based TTA methods.** A large strand of TTA uses teacher–student consistency with an EMA teacher, often improving stability in non-stationary streams (e.g., CoTTA[1], RMT[2], AR-TTA[3], especially RMT). These methods are commonly viewed as less sensitive to within-batch composition because the teacher evolves smoothly via EMA, and predictions used for consistency are augmentation-/time-averaged. CUPLOT does not directly compare against these strong MT baselines, and it’s unclear whether the intra-batch ordering still provides gains once an EMA teacher is present.

[1] Wang, Qin, et al. "Continual test-time domain adaptation." CVPR. 2022.
[2] Döbler, Mario, Robert A. Marsden, and Bin Yang. "Robust mean teacher for continual and gradual test-time adaptation." CVPR. 2023.
[3] Sójka, Damian, et al. "Ar-tta: A simple method for real-world continual test-time adaptation." ICCV. 2023.

**Questions:**

Throughout the manuscript, many citations are written with "\citet", even when the cited work is not the grammatical subject of the sentence. In those cases, "\citep" would be more appropriate and reads more naturally under the ICLR style.

---

> ### Author Response · Authors · 2025-11-21
> **Responses to Reviewer 74MR (Part 1 W1, W2)**
>
> Thank you for your careful review and constructive feedback. We address your concerns below.
>
> > **W1:** “Computational overhead not fully accounted. CUPLOT requires per-instance gradients to rank samples and then multiple updates per batch across curricula, while TTA is very sensitive with compute overhead since it is directly deployed at test time. The appendix varies $K^{t}$ and reports time/accuracy trade-offs, but a clear backward-passes-per-sample accounting and wall-clock time comparison vs. strong baselines is missing in the main text.”
>
> We agree that compute overhead is crucial for TTA deployment, and we clarify that CUPLOT introduces bounded and explicitly controllable extra cost. As you observed, Table 8 and 9 in Appendix A.3 provide time–accuracy trade-offs when varying $K^{t}$. Also, Sec. 4.4 reports the latency and memory overhead against strong baselines, with the full results placed in Appendix Table 5 due to space limitations at submission time. We would like to clarify that the latency reported in Table 5 already refers to the actual wall-clock time (per batch) measured for end-to-end test-time adaptation.
>
> To further enhance clarity, we will revise the main text to explicitly state that the latency in Table 5 corresponds to wall-clock time. We will relocate Table 5 to the main text to provide fully transparent overhead reporting in the revised version. Regarding the backward-passes-per-sample (BPPS), which is defined as the number of backward passes per sample, it is directly connected to the curriculum number $K^t$. Since we use gradient consistency to drive the learning order, each sample requires multiple backward passes to update the model parameters, and BPPS = $ \frac{K^t (K^t-1)}{2}$.
>
>
> > **W2**: "Incremental combination rather than a new principle. Prototypes for pseudo-labels are well-trodden ideas. The novelty mainly lies in the specific gradient-consistency-driven ordering with prototype soft labels."
>
> The novelty of our method lies in enabling the model to progressively absorb fine-grained domain knowledge based on the gradient-consistency–driven curriculum learning strategy. Prototype-based pseudo-labels here serve merely as a means to implement the curriculum, rather than the core contribution. While existing prototype-based TTA methods generate pseudo-labels at the batch level using static prototypes independent of sample ordering, our approach CUPLOT arranges an appropriate learning order for test-time adaptation of the model and updates prototypes after each curriculum step, so that pseudo-labels for later curricula are generated using prototypes that have already incorporated fine-grained domain knowledge from earlier samples. To our knowledge, no prior TTA or pseudo-labeling method has combined gradient-consistency–driven ordering with progressively evolving prototypes to explicitly control the temporal structure of pseudo-label generation in the online adaptation stream.
>
> Furthermore, our theoretical analysis in Section 3.3 formalizes why optimal learning order reduces error propagation, an effect not captured by prior prototype-based approaches, demonstrating that our contribution is not merely an incremental combination of existing ideas but a principled mechanism to integrate fine-grained domain knowledge into learning.

---

> ### Author Response · Authors · 2025-11-21
> **Responses to Reviewer 74MR (Part 2 W3)**
>
> > **W3:**  "Effect sizes are small; significance not emphasized. The gains over comparing methods on the three corruption suites are sub-1% on average. It’s hard to tell its advantages."
>
> The average improvements reported in the main tables reflect mean accuracy over various corruption types, which is a notoriously challenging evaluation setting. Here, we further conducted a Wilcoxon signed-rank test across all corruption types to rigorously assess the significance of the improvements. The results in Table A show that our method CUPLOT consistently outperforms all compared baselines, indicating that the observed gains are statistically significant rather than due to chance. As noted in the submission, BN, TENT and TAST-BN are omitted from Table A for ImageNet-C because the ViT-B32 backbone does not contain batch-normalization layers.
>
> **Table A**. Summary of the Wilcoxon signed-ranks test for CUPLOT against comparing approaches at 0.05 significance level. The p-values are shown in the brackets.
>
> |Methods|CIFAR-10-C|CIFAR-100-C|ImageNet-C|
> |---|---|---|---|
> |ERM|win[6.1e-05]|win[6.1e-05]|win[6.1e-05]|
> |BN|win[6.5e-04]|win[6.1e-05]|-|
> |TENT|win[3.1e-04]|win[6.1e-05]|-|
> |PL|win[6.5e-04]|win[6.1e-05]|win[6.1e-05]|
> |SHOT-IM|win[6.1e-05]|win[1.8e-04]|win[6.5e-04]|
> |T3A|win[6.1e-05]|win[6.1e-05]|win[6.1e-05]|
> |TAST|win[6.1e-05]|win[6.1e-05]|win[6.1e-05]|
> |TAST-BN|win[2.2e-03]|win[6.1e-05]|-|
> |TSD|win[1.5e-03]|win[8.0e-04]|win[6.1e-05]|
> |PROGRAM|win[6.1e-05]|win[6.1e-05]|win[6.1e-05]|
> |DEYO|win[6.1e-05]|win[6.5e-04]|win[6.1e-05]|
>
>
> More importantly, CUPLOT’s contribution lies not only in accuracy gains but also in its ability to progressively absorb fine-grained domain knowledge, theoretically reduce error propagation during test-time adaptation, and provide a principled mechanism for stable online adaptation. These benefits are critical for practical robustness.

---

> ### Author Response · Authors · 2025-11-21
> **Responses to Reviewer 74MR (Part 3 W4, Q)**
>
> > **W4**: "Limited comparison to strong Mean-teacher-based TTA methods. A large strand of TTA uses teacher–student consistency with an EMA teacher, often improving stability in non-stationary streams (e.g., CoTTA[1], RMT[2], AR-TTA[3], especially RMT). These methods are commonly viewed as less sensitive to within-batch composition because the teacher evolves smoothly via EMA, and predictions used for consistency are augmentation-/time-averaged. CUPLOT does not directly compare against these strong MT baselines, and it’s unclear whether the intra-batch ordering still provides gains once an EMA teacher is present."
>
> We appreciate the reviewer’s suggestion regarding strong Mean-Teacher (MT)–based TTA methods. MT approaches such as CoTTA[1], RMT[2], and AR-TTA[3] indeed leverage EMA teachers to improve temporal stability in non-stationary test streams. Our goal in CUPLOT is orthogonal to MT techniques: rather than smoothing predictions through EMA, we focus on controlling the intra-stream learning order to enable finer absorption of domain knowledge and to reduce error propagation. The two mechanisms, EMA-based smoothing and curriculum-based ordering, are complementary rather than mutually exclusive.
>
> We conducted additional experiments where we integrated an EMA teacher into our framework. The results in Table B show that the curriculum-driven ordering in CUPLOT continues to provide consistent improvements even when an EMA teacher is present, indicating that the benefits of ordering do not vanish under MT-style stabilization. These MT comparisons will be included in the revised version.
>
> **Table B**. Classification **error rate** (%) on the CIFAR100-to-CIFAR100C online continual test-time adaptation task under the highest corruption severity level (severity 5), evaluated on ResNeXt-29 following the RMT protocol.
>
> |Methods|Gaussian|shot|impulse|defocus|glass|motion|zoom|snow|frost|fog|brightness|contrast|elastic|pixelate|jpeg|Mean|
> |---|---|---|---|---|---|---|---|---|---|---|---|---|---|---|---|---|
> |Source|73.0|68.0|39.4|29.3|54.1|30.8|28.9|39.5|45.8|50.3|29.5|55.1|37.2|74.7|41.2|46.4|
> |BN|42.1|40.7|42.7|27.6|41.9|29.7|27.9|34.9|35.0|41.5|26.5|30.3|35.7|32.9|41.2|35.4|
> |TENT|37.2|35.8|41.7|37.9|51.2|48.3|45.8|58.4|63.7|71.1|70.4|82.3|88.0|88.5|90.4|60.9|
> |CoTTA|40.1|37.7|39.7|26.9|38.0|27.9|26.4|32.8|31.8|40.3|24.7|26.9|32.5|28.3|33.5|32.5|
> |RMT|38.5|34.4|35.4|26.4|32.7|27.0|25.0|27.6|27.5|30.0|24.0|25.8|27.0|25.2|28.4|29.0|
> |Ours|38.0|34.0|34.8|26.0|32.2|26.4|24.5|27.2|27.1|29.5|23.5|25.3|26.6|24.7|27.9|28.5|
>
> We agree that including MT baselines aimed at non-stationary test streams strengthens the evaluation. Hence, we will further add the following discussion in Related Work:
>
> *"A complementary line of work [1,2,3], especially the effective approach RMT[2], focuses on the setting of Continual Test-Time Adaptation (CoTTA) and adopts EMA-based teacher–student consistency, which provides highly stable adaptation in non-stationary environments through teacher–student consistency."*
>
> [1] Wang, Qin, et al. "Continual test-time domain adaptation." CVPR. 2022.
> [2] Döbler, Mario, Robert A. Marsden, and Bin Yang. "Robust mean teacher for continual and gradual test-time adaptation." CVPR. 2023.
> [3] Sójka, Damian, et al. "Ar-tta: A simple method for real-world continual test-time adaptation." ICCV. 2023.
>
>
> > **Q1:** "Throughout the manuscript, many citations are written with "\citet", even when the cited work is not the grammatical subject of the sentence. In those cases, "\citep" would be more appropriate and reads more naturally under the ICLR style."
>
> We appreciate the reviewer’s careful attention to the manuscript’s writing. We agree that in several places the use of `\citet` is not stylistically appropriate under ICLR conventions. We will carefully revise all in-text citations and replace these occurrences with `\citep` when the cited work is not the grammatical subject of the sentence. The revised version will fully comply with the official ICLR citation style.
>
>
> Thank you again for your valuable suggestions.

---

> ### Author Response · Authors · 2025-11-27
>
> Dear Reviewer 74MR,
>
> We greatly appreciate the time and effort you've taken to review our submission. We hope that our response has addressed the concerns raised in your initial reviews, and we look forward to your feedback.
>
> Please let us know if you need any further information or clarification.
>
> Best regards and thanks,
>
> Paper 15203 Authors

---

### Author Response · Authors · 2025-11-21
**Revision Uploaded**

We thank all reviewers for their constructive feedback and have updated our paper accordingly. Please take a moment to check out the revised manuscript, which incorporates the following key modifications (highlighted in blue):

**New Results**
- **Visuals**: learning-order comparison curves (Figure 1(a)) and a framework diagram (Figure 1(b)), which strengthen the motivation and illustrate the key components of our approach.
- **Significance tests**: added Wilcoxon signed-rank tests across corrupted benchmark datasets (Table 1, Section 4.3).
- **Compute accounting**: moved wall-clock latency table to main text (Table 5) and added backward-passes-per-sample (BPPS) metric (Table 5, Section 4.4).
- **Mean-Teacher baselines**: new experiments under Continual Test-Time Adaptation vs. CoTTA[1], RMT[2]; CUPLOT could extend to non-stationary settings while maintaining competence. (Section 4.5, Appendix A.3, Table 6).
- **Alternative difficulty metrics**: added entropy-confidence and prototype-distance baselines; gradient consistency outperforms them (Appendix A.8).
- **Sequence ablation**: random / sequential order vs. curriculum; curriculum beats both by ≥0.8 % (Appendix A.9, Table 15).
- **Plug-and-play study**: integrated CUPLOT curriculum into SHOT[3] and DeYO[4]—consistent +0.5–1.2% gains (Appendix A.10, Table 16).


**Other Revisions**
- **Related work** re-structured into (i) problem setting, (ii) adaptation methods, including most recent works in Continual Test-Time Adaptation (iii) curriculum learning (Section 2).
- **Theory** clarified: assumptions are standard Lipschitz bounds; Theorem 1 now explicitly states faster convergence(Section 3.3).
- **Preliminary** extended: connections to curriculum learning are added.
- **Notation** polished: key equations re-typeset for clarity; `\citep` used consistently per ICLR style.

We appreciate the great efforts by the AC and reviewers. Please let us know if any further comments arise.

[1] Wang et al. Continual test-time domain adaptation. CVPR, 2022.

[2] Döbler et al. Robust mean teacher for continual and gradual test-time adaptation. CVPR. 2023.

[3] Liang et al. Do we really need to access the source data? source hypothesis transfer for unsupervised domain adaptation. ICML 2020.

[4] Lee et al. Entropy is not enough for test-time adaptation: From the perspective of disentangled factors. ICLR, 2024.

---

### Meta-Review · Area_Chair_SQKE · 2025-12-28

**Summary:**

74MR: (1) computational overhead not fully accounted. (2) limited novelty. (3) limited gain over compared methods. (4) limited comparison to strong mean-teacher-based TTA methods.

pezg: (1) limited experimental scope. (2) some writing issues (Introduction and Method, related work)

tWX1: (1) need more experiments or theoretical analysis for the motivation. (2) need a comparison between different sequence selection strategies. (3) some issues related to presentation/clarify

9R4p: (1) need solid experimental or theoretical foundation for the motivation. (2) need justification for the gradient-based curriculum selection. (3) the theoretical analysis is not rigorous. (4) need experiments on continual TTA setting. (5) need clearer and more intuitive explanation of the proposed method.

This paper got mixed ratings. The main concerns raised by the reviewers include limited novelty, limited performance gain, presentation/clarity, more experiment/theoretical justification, etc. Although the rebuttal addressed some of the concerns, a lot of the other concerns are still outstanding. Given these, the paper is not ready for ICLR. Authors are encouraged to take into account of reviewers' comments and revise the paper for resubmission.

**Reviewer Concerns:**

Reviewer 9R4p explicitly mentioned that most of his/her concerns (especially on theoretical explanations) are addressed. Other reviewers have not responded.

The AC had a look at the reviews and rebuttal. Other concerns (e.g. limited novelty, limited performance gain, clarity, etc) are not well addressed in the rebuttal.

**Reviewer Scores:**

Reviewer 9R4p explicitly mentioned that he/she will increase the score. For other reviewers, they are not very likely to change their scores.

---

### Decision · Program_Chairs · 2026-01-26

Reject